# The Disagreement Problem in Explainable Machine Learning: A Practitioner's Perspective

**Satyapriya Krishna⋆**                                               *skrishna@g.harvard.edu*
*Harvard University*

**Tessa Han⋆**                                                         *than@g.harvard.edu*
*Harvard University*

**Alex Gu**                                                            *gua@mit.edu*
*Massachusetts Institute of Technology*

**Steven Wu**                                                          *zstevenwu@cmu.edu*
*Carnegie Mellon University*

**Shahin Jabbari**                                                     *shahin@drexel.edu*
*Drexel University*

**Himabindu Lakkaraju**                                                *hlakkaraju@hbs.edu*
*Harvard University*

⋆ These authors contributed equally to this work.

**Reviewed on OpenReview:** `https://openreview.net/forum?id=jESY2WTZCe`

## Abstract

As various post hoc explanation methods are increasingly being leveraged to explain complex models in high-stakes settings, it becomes critical to develop a deeper understanding of whether and when the explanations output by these methods disagree with each other, and how such disagreements are resolved in practice. However, there is little to no research that provides answers to these critical questions. In this work, we formalize and study the disagreement problem in explainable machine learning. More specifically, we define the notion of disagreement between explanations, analyze how often such disagreements occur in practice, and how practitioners resolve these disagreements. We first conduct interviews with data scientists to understand what constitutes disagreement between explanations generated by different methods for the same model prediction, and introduce a novel quantitative framework to formalize this understanding. We then leverage this framework to carry out a rigorous empirical analysis with four real-world datasets, six state-of-the-art post hoc explanation methods, and six different predictive models, to measure the extent of disagreement between the explanations generated by various popular explanation methods. In addition, we carry out an online user study with data scientists to understand how they resolve the aforementioned disagreements. Our results indicate that (1) state-of-the-art explanation methods often disagree in terms of the explanations they output, and (2) machine learning practitioners often employ ad hoc heuristics when resolving such disagreements. These findings suggest that practitioners may be relying on misleading explanations when making consequential decisions. They also underscore the importance of developing principled frameworks for effectively evaluating and comparing explanations output by various explanation techniques.

## 1 Introduction

As machine learning (ML) models become integral in high-stakes domains like healthcare and finance, the interpretability of these models is crucial for ML practitioners and domain experts (e.g., doctors and policymakers). Understanding model behavior is essential to identify errors and biases and gauge the reliability of model predictions, especially in complex models (Doshi-Velez & Kim, 2017). Several post hoc explanation methods have been proposed such as perturbation-based methods (e.g., LIME (Ribeiro et al., 2016b), SHAP (Lundberg & Lee, 2017)) and gradient-based methods (e.g., Gradient times Input (Simonyan et al., 2014), SmoothGrad (Smilkov et al., 2017), Integrated Gradients (Sundararajan et al., 2017), GradCAM (Selvaraju et al., 2017)).

Prior research has investigated various aspects of explanation methods. For example, metrics have been developed to assess the faithfulness of explanations to the model's behavior, such as comparing them to a model's ground truth feature importances or using the "Remove and Retrain" (ROAR) approach (Liu et al., 2021; Hooker et al., 2018). However, challenges arise due to the unavailability of ground truth in real-world applications and the impracticality of retraining models in certain settings (Zhou et al., 2021). Studies have shown that popular explanation methods can produce inconsistent or misleading explanations, leading to risks like deploying biased models or making erroneous clinical decisions (Aivodji et al., 2019; Slack et al., 2020).

However, while prior research has studied explanation methods, the (dis)agreement among different methods has not been extensively investigated. Practitioners often use multiple methods, but these can yield conflicting explanations. Therefore, resolving these disagreements is crucial to avoid reliance on misleading explanations with potentially severe consequences (Slack et al., 2020). Yet, there is a lack of research on the extent to which explanations disagree and how practitioners resolve these disagreements.

In this work, we formalize and investigate the *disagreement problem* in explainable ML, a novel area of research that examines conflicts among post hoc explanation methods. We present the following key contributions:

1. We conduct semi-structured interviews with 25 data scientists, gaining insights on the nature and frequency of explanation disagreements in their workflows. This input helps us define what constitutes an explanation disagreement in practical scenarios. [1].

2. Building on these insights, we formalize the concept of explanation disagreement and create a new evaluation framework to quantitatively assess disagreements between pairs of explanations for the same model prediction.

3. We apply this framework in an extensive empirical analysis, measuring disagreement levels across four real-world datasets, three data modalities, six advanced explanation methods, and various predictive models, including logistic regression, tree-based models, and various neural networks.

4. Finally, we conduct an online user study with another set of 25 data scientists. We present them with pairs of conflicting explanations, assessing which they would rely on and their rationale. We also gather insights on their strategies for resolving such disagreements in their daily work.

Our findings reveal that explanation disagreements are common and problematic: 84% of interviewees reported encountering explanation disagreements in practice. Our empirical analysis confirms widespread discrepancies across different methods and data modalities (tabular, text, and image). Moreover, 86% of our online user study participants admitted to using ad hoc heuristics or being uncertain about resolving explanation disagreements. This highlights the need for better evaluation metrics and enhanced practitioner understanding of the underlying principles of explanation methods.

## 2 Related Work

Our work is grounded in the extensive literature on explainable ML, which we discuss in this section.

---

[1]All the user interviews and studies in this work were approved by our institution's IRB.

**Inherently Interpretable Models and Post Hoc Explanations.** A variety of approaches have been developed to learn inherently interpretable models for tasks such as classification and clustering (Letham et al., 2015; Lakkaraju et al., 2016; Bien & Tibshirani, 2009; Kim et al., 2014; Lou et al., 2012; Caruana et al., 2015). However, complex models like deep neural networks often outperform these simpler models in terms of accuracy (Ribeiro et al., 2016b). This has led to significant interest in post hoc explanation methods, which aim to understand the behavior of these complex models by approximating them using simpler models in a post hoc fashion. Post hoc explanation methods can be categorized into local explanation methods and global explanation methods. Local explanation methods, including LIME, SHAP, SmoothGrad, and Integrated Gradients (Ribeiro et al., 2016a; Lundberg & Lee, 2017; Sundararajan et al., 2017; Smilkov et al., 2017) focus on explaining individual model predictions.

Global explanation methods, on the other hand, aim to summarize the behavior of complex models as a whole (Bastani et al., 2017; Lakkaraju et al., 2019). Our work is focused on analyzing the disagreements among explanations generated by the aforementioned local post hoc explanation methods.

**Analyzing and Evaluating Post hoc Explanations.** Prior research has proposed several notions of explanation quality, such as faithfulness, stability, consistency, and sparsity (Liu et al., 2021; Petsiuk et al., 2018; Slack et al., 2021; Zhou et al., 2021). Metrics have also been proposed to quantify these aspects of explanation quality (Zhou et al., 2021; Carvalho et al., 2019; Gilpin et al., 2018; Liu et al., 2021). These properties and metrics have been leveraged to theoretically and empirically analyze the behavior of popular post hoc explanations (Ghorbani et al., 2019; Slack et al., 2020; Dombrowski et al., 2019; Adebayo et al., 2018; Alvarez-Melis & Jaakkola, 2018; Levine et al., 2019; Chalasani et al., 2020; Agarwal et al., 2021). The work that is closest to ours is the research by Neely et al. (2021) which demonstrates that certain post hoc explanation methods (e.g., LIME, Integrated Gradients, DeepLIFT, Grad-SHAP, Deep-SHAP, and attention-based explanations) disagree with each other based on rank correlation (Kendall's $\tau$) metric. However, their work neither formalizes the notion of explanation disagreement by leveraging practitioner inputs nor studies how explanation disagreements are resolved in practice, which are the key contributions of this work. This research has also inspired subsequent works that delve deeper into this topic (Han et al., 2022; Banegas-Luna et al., 2023). For instance, Han et al. (2022) built on this work and theoretically analyzed multiple state-of-the-art explanation methods to understand the reasons behind their disagreements.

**Human Factors in Explainability.** Many user studies have been conducted to evaluate how well humans can understand and use explanations (Doshi-Velez & Kim, 2017). Some of these studies have shown that data scientists often struggle to understand and effectively leverage state-of-the-art explanation techniques (Kaur et al., 2020). Others have identified a variety of stakeholders across the model life cycle and highlighted the core goals of model explanations (Hong et al., 2020; Chen et al., 2022). Prior works have also examined whether explanations help users in performing specific tasks (Buçinca et al., 2021; Bansal et al., 2021; Fan et al., 2022; Sperrle et al., 2021; Chen et al., 2022; Vasconcelos et al., 2023; Fok & Weld, 2023). However, none of these works focus on understanding the disagreement problem, the extent to which practitioners face it, and/or how they resolve it.

## 3 Understanding and Measuring Disagreement Between Model Explanations

In this section, we discuss practitioner perspectives on what constitutes disagreement between two explanations and then formalize the notion of explanation disagreement. To this end, we first describe the study that we carry out with data scientists to understand what constitutes explanation disagreement, and the extent to which they encounter this problem in practice. We then discuss the insights from this study and leverage these insights to propose a novel framework that can quantitatively measure the disagreement between any two explanations.

### 3.1 Characterizing Explanation Disagreement Using Practitioner Inputs

Here, we describe the interviews we conducted with data scientists to characterize explanation disagreement and outline findings and insights from these interviews.

### 3.1.1 Interviews with Data Scientists

We conducted 30-minute semi-structured interviews with 25 data scientists who employ explainability techniques to understand model behavior and explain it to their customers and managers. We recruited these participants by emailing the explainable machine learning groups (and relevant mailing lists) in three different for-profit companies in the technology and financial services sectors across the United States. We conducted basic checks (e.g., pre-screening interviews and profile checks) to ensure that each recruited participant has at least one year of experience as a data scientist, and has working knowledge of data science and machine learning.

Furthermore, all of these data scientists used state-of-the-art local post hoc explanation methods (e.g., LIME, SHAP, and gradient-based methods) in their day-to-day workflow. 19 participants (76%) were male, and 6 (24%) were female. 16 participants (64%) had more than 2 years of experience working with explainability techniques, and the remaining 9 (36%) had about 8 to 12 months of experience. Our interviews included, but were not limited to the following questions: Q1) *How often do you use multiple explanation methods to understand the same model prediction?* Q2) *What constitutes disagreement between two explanations that explain the same model prediction?* Q3) *How often do you encounter disagreements between explanations output by different methods for the same model prediction?*

### 3.1.2 Findings and Insights

The interviews revealed how data scientists use explanation methods and their perspectives on what constitutes disagreement among explanations. 22 out of the 25 participants (88%) said that they almost always use multiple explanation methods to understand the same model prediction. Furthermore, 21 out of the 25 participants (84%) mentioned that they have often run into some form of disagreement between explanations generated by different methods for the same prediction. We found that data scientists consider explanations to disagree when:

**Top features are different:**  Most popular post hoc explanation methods return feature importance values for each feature. These values indicate which features contribute most (either positively or negatively) to the prediction, i.e., top features. 21 out of the 25 participants (84%) mentioned that such a set of top features is *"the most critical piece of information"* that they rely on in their day-to-day workflow. They also noted that they typically look at the top 5 to 10 features provided by an explanation for each prediction. When two explanations have different sets of top features, they consider it to be a disagreement.

**Ordering among top features is different:**  18 out of 25 participants (72%) indicated that they also consider the ordering among the top features very carefully in their workflow. Therefore, they consider a mismatch in the ordering of the top features provided by two different explanations to be a disagreement.

**Direction of top feature contributions is different:**  19 out of 25 participants (76%) mentioned that the *sign* or *direction* of the feature contribution (i.e., whether the feature contributes positively or negatively to the prediction) is another critical piece of information. Any mismatch in the signs of the top features between two explanations is a sign of disagreement. As one participant remarked: *"I saw an explanation indicating that a top feature bankruptcy contributes positively to a particular loan denial, and another explanation saying that it contributes negatively. That is a clear disagreement. The model prediction can be trusted with the former explanation, but not with the latter."*.

**Relative ordering of features of interest is different:**  16 of our study participants (64%) indicated that they also look at relative ordering between certain features of interest, and that, they consider explanations to disagree if explanations provide contradicting information about this aspect. As one participant remarked: *"I often check if salary is more important than credit score in loan approvals. If one explanation says salary is more important than credit score, and another says credit score is more important than salary; then it is a disagreement."*

A striking finding from our study is that participants characterize explanation disagreement based on factors such as mismatch in top features, feature ordering, directions of feature contributions, and relative feature ordering, but not on the feature importance values output by different explanation methods. 24 out of 25 participants (96%) opine that feature importance values output by different explanation methods are not directly comparable. For example, they note that while LIME outputs coefficients of a linear model as feature importance values, SHAP outputs Shapley values as feature attributions which sum to the probability of the predicted class. Thus, data scientists do not try to base explanation disagreement on these numbers not being equal or similar. One of our participants succinctly summarized practitioners' perspectives on this explanation disagreement problem: *"The values generated by different explanation methods are different. So, I would not characterize disagreement based on that. But, I would at least want the explanations they output to give me consistent insights. The explanations should agree on what are the most important features, the ordering among them, and so on for me to derive consistent insights. But, they don't!"*

### 3.2 Formalizing the Notion of Explanation Disagreement

Our study indicates that ML practitioners consider the following key aspects when they think about explanation disagreement: a) the extent to which explanations differ in the top-$k$ features, the signs of these top-$k$ features, and the ordering of these top-$k$ features, and b) the extent to which explanations differ in the relative ordering of certain features of interest. To capture these intuitions about explanation disagreement, we propose six different metrics: *feature agreement*, *rank agreement*, *sign agreement*, *signed rank agreement*, *rank correlation*, and *pairwise rank agreement*. The first four metrics capture disagreement with respect to the top-$k$ features of the explanations and the last two metrics capture disagreement with respect to a selected set of features that could be provided as input by an end user. For all six metrics, lower values indicate stronger disagreement. See Section A for formal definitions.

#### 3.2.1 Measuring Disagreement With Respect to Top-k Features

We now define four metrics, which capture specific aspects of explanation disagreement with respect to the top-$k$ features.[2]

**Feature Agreement:** ML practitioners in interviews (Section 3.1) indicated that a key notion of disagreement between a pair of explanations is that they output different top-$k$ features. To capture this notion, feature agreement computes the fraction of common features between the sets of top-$k$ features of the two explanations.

**Rank Agreement:** Practitioners also indicated that if the ordering of the top-$k$ features is different for two explanations (even if the feature sets are the same), then they consider it to be a disagreement. To capture this notion, rank agreement computes the fraction of features that are not only common between the sets of top-$k$ features of two explanations, but also have the same position in the respective rank orders. Rank agreement is a stricter metric than feature agreement since it also considers the ordering of the top-$k$ features. While the practitioners we interviewed favored the aforementioned definition of the rank agreement metric, one could also imagine a less strict variant that accounts for the differences between the ranks of the top-k features in the two explanations. Such a *weighted rank agreement* metric would quantify the degree of agreement based on how far apart the top-k features are, rather than only considering an exact match of their ranks. This implies that even if certain features are not in the same position, their relative closeness still contributes positively to the overall agreement score, making it a softer variant of the rank agreement defined above. See Section D.2 for more details on weighted rank agreement and additional disagreement metrics that we explored.

**Sign Agreement:** In our study, practitioners also mentioned that they consider two explanations to disagree if the feature attribution signs (i.e. directions of feature contribution) do not align for the top-$k$ features. To capture this notion, sign agreement computes the fraction of features that are not only common

---

[2]The top-$k$ features of an explanation are typically computed only based on the magnitude of the feature importance values and not the signs.

between the sets of top-$k$ features of two explanations, but also share the same sign in both explanations. Sign agreement is a stricter metric than feature agreement since it also considers signs of the top-$k$ features.

**Signed Rank Agreement:** This metric fuses the above three notions of explanation disagreement and computes the fraction of features that are not only common between the sets of top-$k$ features of two explanations but also share the same feature attribution sign and rank in both explanations. The signed rank agreement is the strictest compared to all the aforementioned metrics since it considers both the ordering and the signs of the top-$k$ features.

### 3.2.2 Measuring Disagreement With Respect to Features of Interest

Practitioners also indicated that they consider two explanations to be different if the relative ordering of features of interest (e.g., salary and credit score discussed in Section 3.1) differ between the explanations. These features could be provided as input by an end user (such as a data scientist). To formalize this notion, we introduce the two metrics below.

**Rank Correlation:** In practice, this selected set of features corresponds to features that are of interest to end users and can be provided as input by end users. Rank correlation computes Spearman's rank correlation coefficient between feature rankings provided by two explanations for these selected sets of features.

**Pairwise Rank Agreement:** The pairwise rank agreement takes as input a set of features that are of interest to the user and computes the fraction of feature pairs for which the relative ordering is the same between two explanations.

## 4 Empirical Analysis of Explanation Disagreement

We leverage the metrics outlined in Section 3 and carry out a comprehensive empirical analysis with six state-of-the-art explanation methods and four real-world datasets spanning three data modalities to study the explanation disagreement problem. In this section, we describe the datasets, experimental setup, and key findings.

### 4.1 Datasets

To carry out our empirical analysis, we use four datasets spanning three data modalities (tabular, text, and image). For **tabular** data, we use the Correctional Offender Management Profiling for Alternative Sanctions (COMPAS) (ProPublica) and German Credit datasets (Repository). The COMPAS dataset comprises seven features on the demographics, criminal history, and prison time of 4,937 defendants. Each defendant is classified as either low or high risk for recidivism based on the COMPAS algorithm's risk score. The German Credit dataset contains twenty features on the demographics, credit history, bank account balance, loan information, and employment information of 1,000 loan applicants. Each applicant is classified as either low or high risk for defaulting on a loan. For **text** data, we use Antonio Gulli (AG)'s corpus of news articles (AG_News) (ag, 2005). It contains 127,600 sentences (collected from 1,000,000+ articles from 2,000+ sources with a vocabulary size of 95,000+ words). The class label is the topic of the article from which a sentence was obtained (World, Sports, Business, or Science/Technology). For **image** data, we use the ImageNet$-1k$ (Russakovsky et al., 2015; ima, 2015) object recognition dataset. It contains 1,381,167 images belonging to 1000 categories. We obtain segmentation maps from PASCAL VOC 2012 (voc, 2012) that are directly used as super-pixels for the explanation methods.

### 4.2 Experimental Setup

We train a variety of black box models. For tabular data, we train four models: logistic regression, densely connected feed-forward neural network, random forest, and gradient-boosted tree. For text data, we train a widely-used vanilla LSTM-based text classifier on AG_News (Zhang et al., 2015) corpus. For image data, we use the pre-trained ResNet-18 (He et al., 2016) for ImageNet.

Next, we apply six state-of-the-art post hoc explanation methods to explain the black box models' predictions for a set of test data points.[3] We apply two perturbation-based explanation methods (LIME (Ribeiro et al., 2016b) and KernelShap (Lundberg & Lee, 2017)), and four gradient-based explanation methods (Vanilla Gradient (Simonyan et al., 2014), Gradient times Input (Shrikumar et al., 2017), Integrated Gradients (Sundararajan et al., 2017), and SmoothGrad (Smilkov et al., 2017)). For explanation methods with a sample size hyper-parameter, we either run the explanation method to convergence (i.e., select a sample size such that an increase in the number of samples does not significantly change the explanations) or use a sample size that is much higher than the sample size recommended by previous work.

We then evaluate the (dis)agreement between the explanation methods using the metrics described in Section 3.2. For tabular and text data, we apply rank correlation and pairwise rank agreement across all features; and feature agreement, rank agreement, sign agreement, and signed rank agreement across top-$k$ features for varying values of $k$. For image data, metrics that operate on the top-$k$ features are more applicable to super-pixels. Thus, we apply the six disagreement metrics on explanations output by LIME and KernelSHAP (which leverage superpixels), and calculate rank correlation (across all pixels as features) between the explanations output by gradient-based methods. Additional details are described in Appendix B.

### 4.3 Results and Insights

We discuss the results of our empirical analysis for each of the three data modalities.

### 4.3.1 Tabular Data

Figure 1 shows the disagreement between various pairs of explanation methods for the neural network model trained on the COMPAS dataset. We computed the six metrics outlined in Section 3.2 using $k = 5$ (out of 7 features) for metrics that focus on top-$k$ features. Each cell in the heatmap shows the metric value averaged over the test data points for each pair of explanation methods, with lighter colors indicating stronger disagreement. We see that explanation methods tend to exhibit slightly higher values on pairwise rank agreement and feature agreement metrics and relatively lower values on other metrics (indicating stronger disagreement).

We next study the effect of the number of top features on the degree of disagreement. Figure 2 shows the disagreement of explanation methods for the neural network model trained on the COMPAS dataset. We computed the rank agreement (top row) and signed rank agreement (bottom row) at top-$k$ features for increasing values of $k$. We see that as the number of top-$k$ features increases, rank agreement and signed rank agreement decrease. This indicates that, as $k$ increases, top-$k$ features of a pair of explanation methods are less likely to contain shared features with the same rank (as measured by rank agreement) or shared features with the same rank and sign (as measured by signed rank agreement). These patterns are consistent across other models trained on the COMPAS dataset. See Appendix C.1.

In addition, across metrics, values of $k$, and models, the specific explanation method pairs of Vanilla Gradient-SmoothGrad and Gradient times Input-Integrated Gradients exhibit strong agreement, while the pairs of Vanilla Gradient-Integrated Gradients, Vanilla Gradient-Gradient times Input, SmoothGrad-Gradient times Input, and SmoothGrad-Integrated Gradients consistently exhibit stronger disagreement. These results are consistent with the theoretical findings indicating that certain pairs of explanation methods are more consistent with one another than other pairs (Han et al., 2022).

Furthermore, there are varying degrees of disagreement among pairs of explanation methods. For example, for the neural network model trained on the COMPAS dataset, rank correlation displays a wide range of values across explanation method pairs, with 14 out of 15 explanation method pairs even exhibiting negative rank correlation when explaining multiple data points. This is shown in the left panel of Figure 3, which displays the rank correlation over all the features among all pairs of explanation methods for the neural network model trained on the COMPAS dataset.

---

[3] We also studied another explanation method called L2X (Chen et al., 2018). See Section D.1 for more details.

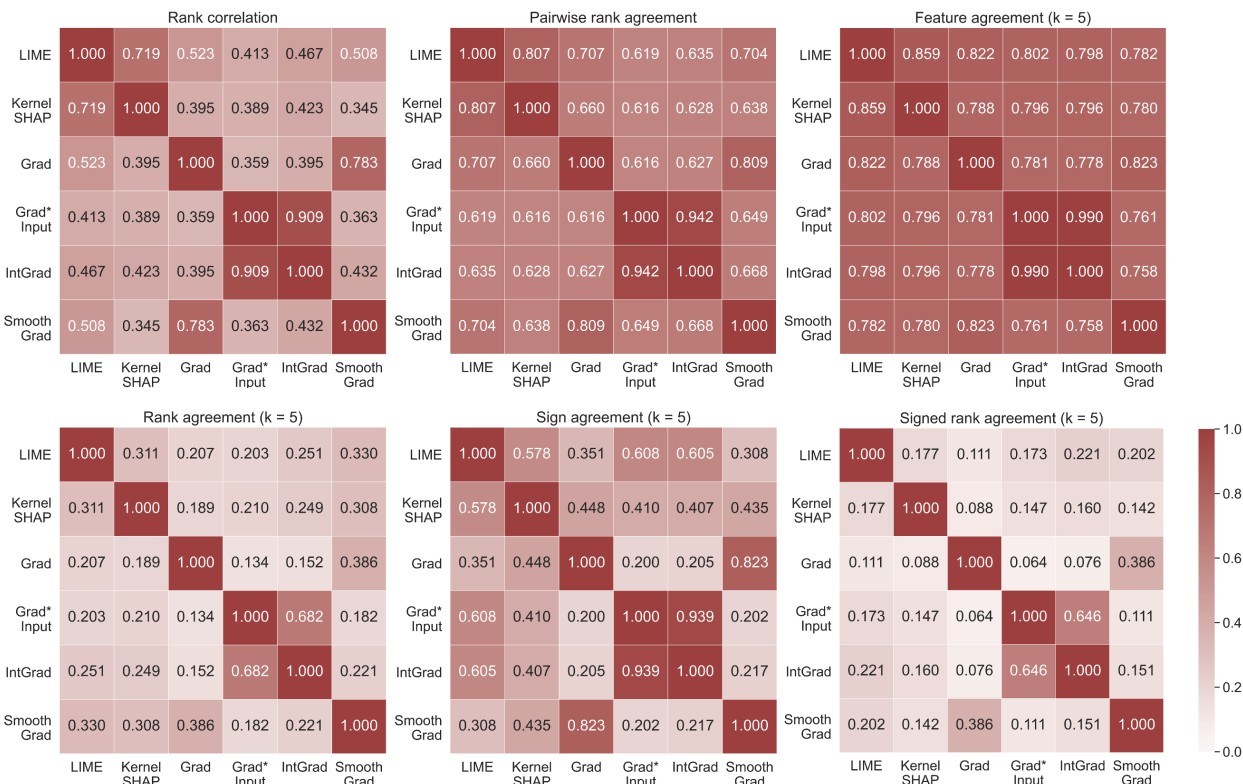

Figure 1: Disagreement between explanation methods for neural network model trained on COMPAS dataset measured by six metrics: rank correlation and pairwise rank agreement across all features, and feature, rank, sign, and signed rank agreement across top $k = 5$ features. Heatmaps show the average metric value over the test set data points for each pair of explanation methods, with lower values (lighter colors) indicating stronger disagreement. The disagreement between a pair of explanations is considered non-trivial if less than 75% of the features agree (i.e., if feature agreement, rank agreement, sign agreement, signed rank agreement, or pairwise rank agreement is less than 0.75), or if the correlation among features is only moderately positive, nonexistent, or negative (i.e. if the rank correlation is less than 0.50). Across all six heatmaps, the standard error ranges between 0 and 0.01.

All the patterns discussed above are also generally reflected in the German Credit dataset (Appendix C.1). However, explanation methods sometimes display stronger disagreement for the German Credit dataset than for the COMPAS dataset. For example, rank agreement and signed rank agreement are lower for the German Credit dataset than for the COMPAS dataset at the top 25%, 50%, 75%, and 100% of features for both logistic regression and neural network models. One possible reason is that the German Credit dataset has a larger set of features than the COMPAS dataset, resulting in a larger number of possible ranking and sign combinations assigned by a given explanation method and making it less likely for two explanation methods to produce consistent explanations.

Moreover, explanation methods display trends associated with model complexity. For example, the disagreement between explanation methods is similar or stronger for the neural network model than for the logistic regression model across metrics and values of $k$ for both COMPAS and German Credit datasets (Appendix C.1). In addition, explanation methods show similar levels of disagreement for the random forest and gradient-boosted tree models. These trends suggest that disagreement among explanation methods may increase with model complexity. As the complexity of the black box model increases, it may be more difficult to accurately approximate the black box model with a simpler model (LIME's strategy, for example) and more difficult to disentangle the contribution of each feature to the model's prediction. Thus, the higher the model complexity, the more difficult it may be for different explanation methods to generate the true

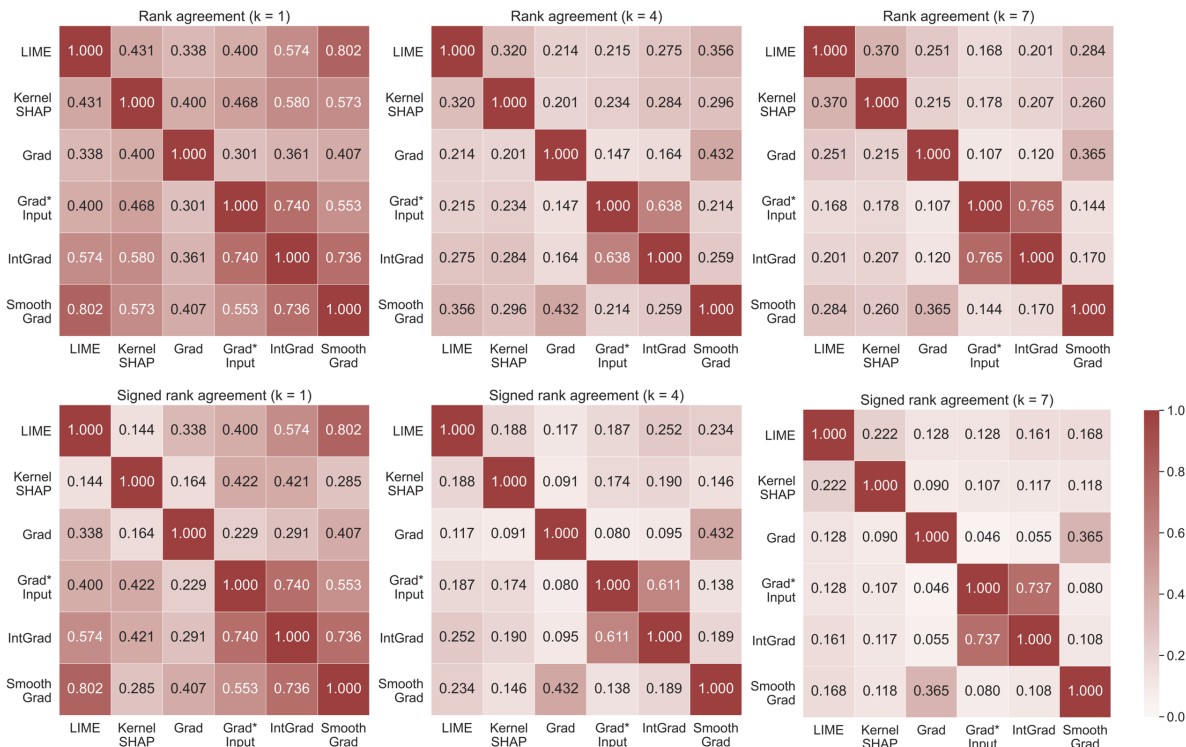

Figure 2: Disagreement between explanation methods for neural network model trained on COMPAS dataset measured by rank agreement (top row) and signed rank agreement (bottom row) at top-$k$ features for increasing values of $k$. Each cell in the heatmap shows the metric value averaged over test set data points for each pair of explanation methods, with lower values (lighter colors) indicating stronger disagreement. We follow the same disagreement interpretation described in Figure 1. Across all six heatmaps, the standard error ranges between 0 and 0.01.

explanation and the more likely it may be for different explanation methods to generate differently false explanations, leading to stronger disagreement among explanation methods. Finally, we also study the effect of dataset complexity (i.e., the number of features) on the degree of disagreement for each of our metrics. See Section D.3 for more details.

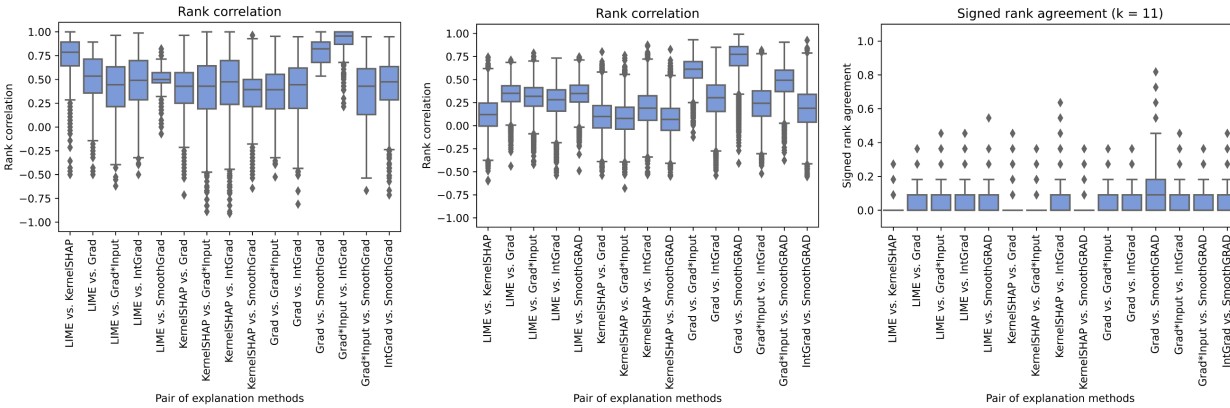

Figure 3: Distribution of rank correlation over all features for neural network model trained on COMPAS (left), and rank correlation across all features (middle) and signed rank agreement across top-11 features (right) for neural network model trained on AG_News.

### 4.3.2 Text Data

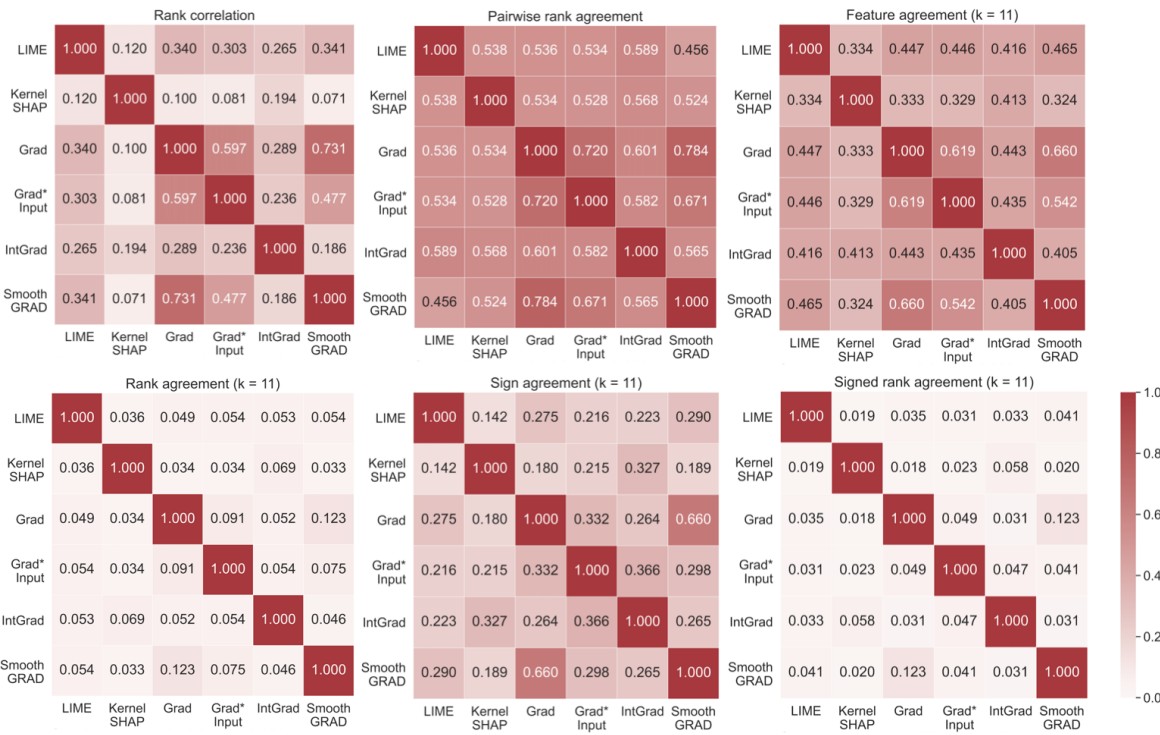

Figure 4: Disagreement between explanation methods for LSTM trained on the AG_News dataset using $k = 11$ features for metrics operating on top-$k$ features and all features for other metrics. Each heatmap shows the metric value averaged over test data for each pair of explanation methods. Lighter colors indicate more disagreement. Standard error ranges from 0.0 to 0.0025 for all six metrics.

In the case of text data, we deal with a high-dimensional feature space where words are features. We plot the six metrics for $k = 11$, which is around 25% of the average text length of a sentence (data point) in the dataset (Figure 4). As seen in the figure, we observe severe disagreements across all six disagreement metrics. Rank agreement and signed rank agreement are the lowest between explanations, with values under 0.1 for most cases, indicating disagreement in over 90% of the top-$k$ features. Trends are quite similar for rank correlation and feature agreement,t with better agreement between gradient-based explanation methods, such as a feature agreement of 0.61 for the Gradient times Input and Vanilla Gradient method pair.

In addition to the broader disagreements between various explanation methods, we notice specific patterns of agreement between certain explanation methods. Based on the middle and right panels of Figure 3, we observe a higher rank correlation between different pairs of gradient-based explanations. We also observe from Figure 4 that LIME exhibits higher agreement with other explanation methods compared to KernelSHAP (e.g., Figure 4 shows that the average rank correlation between LIME and all other explanation methods is 0.273, as opposed to 0.113 in case of KernelSHAP). This finding is consistent with the insights we observed in the case of the COMPAS dataset (Figure 1, top left panel). Lastly, we notice a higher disagreement among explanation methods for text data than for tabular data.

### 4.3.3 Image Data

While LIME and KernelSHAP consider superpixels of images as features, gradient-based methods consider pixels as features. In addition, the notion of top-$k$ features and the metrics we define on top-$k$ features are not semantically meaningful when we consider pixels as features. We compute all six metrics to capture disagreement between explanations output by LIME and KernelSHAP with superpixels as features. We also

compute rank correlation on all the pixels (features) to capture disagreement between explanations output by gradient-based methods.

Unlike earlier trends with tabular and text data, we see higher agreement between KernelSHAP and LIME on all six metrics: rank correlation of 0.8977, pairwise rank agreement of 0.9302, feature agreement of 0.9535, rank agreement of 0.8478, sign agreement of 0.9218, and signed rank agreement of 0.8193.

However, the trends are quite the opposite when we compute rank correlation at pixel-level for gradient-based methods (See Appendix C). For instance, the rank correlation between Integrated Gradients and SmoothGrad is 0.001, indicating strong disagreement. The disagreement is similarly quite strong in the case of other pairs of gradient-based methods. This suggests that disagreement could potentially vary significantly based on the granularity of image representation.

**Remark 1.** *Throughout this section, we carried out a comprehensive empirical analysis with six explanation methods and four real-world datasets spanning three data modalities to study the disagreement problem. While our results generally show a moderate to high degree of disagreement among all pairs of explanations across all data modalities, we observe that the extent of disagreement is slightly lower when we consider variants of the proposed metrics. For example, we explored additional metrics such as weighted rank agreement, top-k pairwise rank agreement, and top-k rank correlation, and we discuss these metrics and corresponding results in Appendix D.2.*

## 5   Resolving the Disagreement Problem in Practice: A Qualitative Study

After establishing the fact that the majority of practitioners observe disagreement between explanations using both qualitative analysis (Section 3) as well as quantitative analysis (Section 4), we try to understand the different ways practitioners tackle this disagreement. To this end, we conduct a qualitative user study targeted towards explainability practitioners. We next describe our user study design and discuss our findings.

### 5.1   User Study Design

In total, 25 participants participated in our study, 13 from academia and 12 from industry. Participants from academia were graduate students and postdoctoral researchers. In contrast, participants from industry were data scientists and ML engineers from three different for-profit companies in the technology and financial services sectors across the United States. We recruited these participants by emailing the explainable ML groups (and relevant mailing lists) within these organizations. 12 participants in this study (48%) also participated in the interviews in Section 3.1.[4] We conducted basic checks (e.g., pre-screening interviews and profile checks) to ensure that each recruited participant has at least one year of experience as a data scientist, and has working knowledge of data science and ML. 20 of these participants indicated that they had used explanation methods in their work in various ways, including doing research, helping clients explain their models, and debugging their own models. Following the setup in Section 4, we asked participants to compare the output of five pairs of explanation methods on the predictions made by the neural network we trained on the COMPAS dataset. We chose the COMPAS dataset because it only has 7 features, making it easy for participants to understand the explanations.

First, the participants are shown an information page explaining the COMPAS risk score binary prediction setting and various explanation methods. We indicate that we trained a neural network to predict the COMPAS risk score (low or high) from the seven COMPAS features. We also give a brief description of each of these seven features to participants and tell them to assume that the criminal defendant's risk of recidivism is correctly predicted to be high risk. In this information page, we also briefly introduce and summarize the six explanation methods we use in the study (LIME, KernelSHAP, Vanilla Gradient, Gradient times Input, SmoothGrad, and Integrated Gradients). Finally, we provide links to the papers describing each method. We include a screenshot of this information page in Appendix E.1.

Next, participants were shown a series of 5 prompts, a sample of which is shown in Figure 5. Each prompt presented two explanations of the neural network model's prediction corresponding to a particular data

---

[4]The remaining 13 participants from the interviews were unavailable during the time of the second study.

point generated using two different explanation methods (e.g., LIME and KernelSHAP in Figure 5). The explanations (and data points) within these prompts were selected to ensure high disagreement between them as per the metrics discussed in prior sections. We displayed the full set of $k = 7$ COMPAS features, showing the feature importance of each feature. The red and blue bars indicate that the feature contributes negatively and positively, respectively, to the model prediction. Participants were first asked the question *"To what extent do you think the two explanations shown above agree or disagree with each other?"* and given four choices: *completely agree, mostly agree, mostly disagree*, and *completely disagree*. If the participants indicated any level of disagreement (i.e., any of the latter 3 choices), we then asked them *"Since you believe that the above explanations disagree (to some extent), which explanation would you rely on?"* and presented them with three choices: the two explanation methods shown and *"it depends"*. Then, they were asked to explain their response. Participants were allowed to take as much time as they needed.

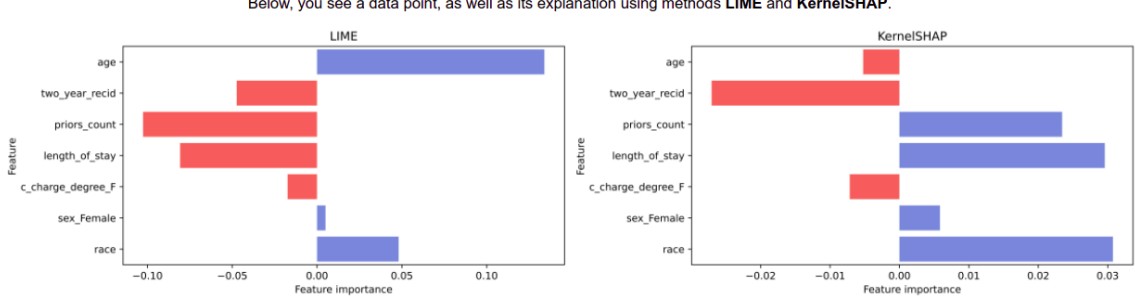

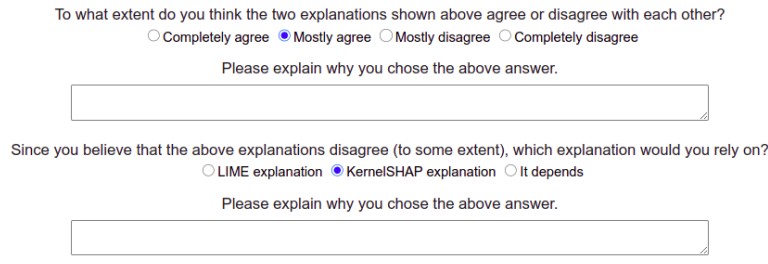

Figure 5: The user interface for a prompt. The user is shown two explanations for a COMPAS data point, showing the feature importance value of each of the 7 features. Red and blue indicate negative and positive feature values, respectively. See the text for more details.

## 5.2 Results and Insights

We now discuss the results and findings from our user study.

### 5.2.1 Do practitioners observe disagreements?

We aggregated the responses to the first question in each prompt, *"To what extent do you think the explanations shown above agree or disagree with each other?"*. Overall, 4%, 28%, 50%, and 18% of responses indicate *completely agree*, *mostly agree*, *mostly disagree*, and *completely disagree*, respectively, highlighting that there is significant disagreement among our prompts. See Appendix E.4 for more details.

### 5.2.2 Are certain explanations favored over others?

Next, since different explanation methods have different levels of popularity, we analyze if certain methods are chosen more often in disagreements. Figure 6a shows the distribution of how participants resolved disagreements for each prompt (dropping prompts with 4 or fewer responses). We first emphasize that there is high variability in how participants chose to resolve disagreements, showing a lack of consensus for the

majority of prompts. However, when participants do decide to choose a method rather than abstaining, they often choose the same method. For example, for the Vanilla Gradient vs. SmoothGrad pair (top row in Figure 6a), participants either chose SmoothGrad over Vanilla Gradient or chose neither. We also aggregate these choices' overall prompts, and in Figure 6b, we plot how often each of the six explanation methods is chosen, finding that indeed, certain methods were favored over others. While KernelSHAP was chosen 66.7% of the time when there were disagreements, Gradient times Input was only chosen 7.0% of the time. We include a further explanation of why participants chose each of the explanations in Appendix E.5, including quotes from participants that supported each method.

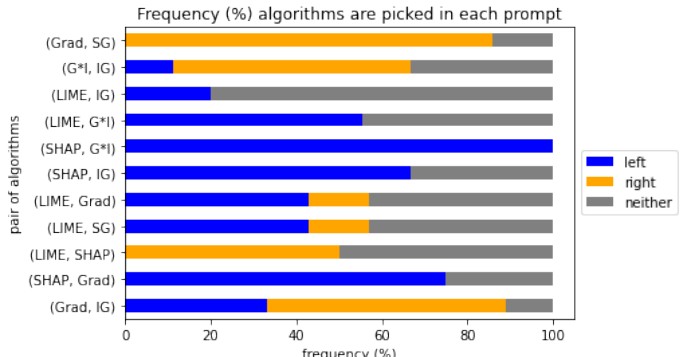
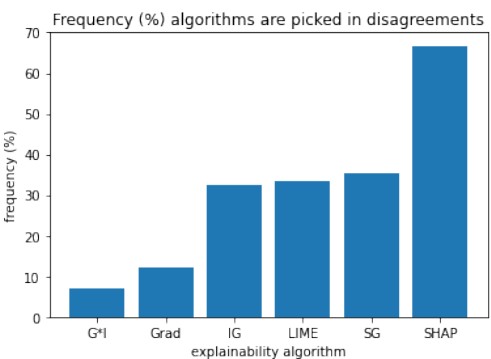

(a) The frequency with which each of the explanations in a pair is selected upon disagreement. The blue, gold, and grey bars show the percentage of participants (X-axis) that picked the left, right, and neither method when presented with the pair of methods shown on the Y-axis.

(b) The frequency with which each of the explanations was chosen when there is a disagreement. X-axis indicates the explanation method and Y-axis indicates the frequency.

Figure 6: Figures highlight which methods participants chose when the explanations they were shown disagreed. In (a), we show how participants resolved each particular prompt. In (b), we show the overall frequencies with which an explanation method was selected.

### 5.2.3 How do practitioners resolve disagreements?

Across all six explanation methods, we find three unifying themes that dictated why participants chose one explanation over the other. We give a high-level description of these themes below, highlighting direct quotes from participants in Table 1.

*1. One method is inherently better than the other because of its associated theory or publication time (33%):* Participants often indicated preference towards a particular method without referencing the shown explanation citing features such as the paper's publication time (more recent is better), the theory behind the method, and the method's stability.

*2. One of the generated explanations matches intuition better (32%):* Participants frequently said that one method's explanation aligned with their intuition better, citing the absolute and relative values of specific features as evidence.

*3. LIME and SHAP are better because the COMPAS dataset comprises tabular data (23%):* Participants said that they mainly used LIME and SHAP for tabular data and commonly cited this as their sole reason.

### 5.2.4 Experiencing and resolving disagreements in day-to-day workflow

After answering all 5 prompts, participants were asked a set of questions on their experience with the disagreement problem in their day-to-day work. First, to filter out participants who didn't use explanation methods, we asked: *"Have you used explanation methods in your work before?".* Of the 25 participants, 5 of them indicated that they had not. We asked the other 20 participants further questions to better understand their experience with the disagreement problem. The full set of questions can also be found in Appendix E.3. Having understood what participants look for to determine disagreement in Section 3.1, we next sought to better understand two crucial questions related to the disagreement problem: *(Q1): Do you observe*

| Theme Highlighted | Sample Quotes |
|---|---|
| **1. One method's paper/theory suggests that it's inherently better (33%).** | • *"I have no reason to believe the gradient holds anywhere other than very locally."* 
 • *"[Integrated Gradients is] more rigorous [than SmoothGrad] based on the paper and axioms"* 
 • *"gradient explanations are more unstable"* |
| **2. One explanation matches intuition better (32%).** | • *"seems unlikely that all features contributed to a positive classification"* 
 • *"features such as priors_count and length of stay [are] important for determining"* 
 • *"Gradient*Input only consider[s] sensitive features (age, race) as impactful which could be a sign of a biased underlying data distribution"* |
| **3. LIME/SHAP are better for tabular data (23%).** | • *"I use LIME for structured data"* 
 • *"SHAP is more commonly used [than Vanilla Gradient] for tabular data"* |

Table 1: Themes summarizing how participants decided between explanations when faced with disagreement along with quotes.

*disagreements between explanations output by state-of-the-art methods in your day-to-day workflow?* and *(Q2): How do you resolve such disagreements in your day to day workflow?*.

One of the 20 participants declined to respond to (Q1) and (Q2) because they were not a practitioner. Out of the other 19, 14 participants (74%) responded *"yes"* to (Q1), indicating that they did in fact encounter explanation disagreement in practice. Of the remaining 5 who said they did not, 3 said they had not really paid attention to the issue. We also aimed to uncover how participants dealt with the disagreement problem when it arose in practice (Q2). 14 participants responded affirmatively to (Q1), and their responses to (Q2) can be grouped into 3 categories. 50% favored using ad-hoc heuristics based on personal preferences for choosing which methods to use (*"picking their favorite method"*, *"rules of thumb based on results in papers"*). These heuristics varied among participants and included ease of implementation, accompanying theoretical results, recency of publication, ease of understanding, and documentation of packages. 36% did not indicate any explicit way for resolving these disagreements but instead demonstrated confusion and uncertainty (*"no clear answer to me"*). Several of these responses also indicated the desire for the research community to make progress and help (*"I hope research community can provide some guidance"*). The remaining 14% proposed to use other metrics such as fidelity (*"try and use some metric to measure fidelity"*). See Appendix E.7 for more details including the breakdown of our results for participants in academia and industry.

# 6 Discussion and Conclusion

The main takeaway from our work is the observation that state-of-the-art explanation methods consistently disagree with each other across various datasets and data modalities, as quantified by the metrics we outlined. Our empirical estimates of explanation disagreement closely align with human assessments of disagreements among leading explainable AI methods. Our focus was on highlighting the prevalence of disagreement within the literature on explainable AI and examining how practitioners address these disagreements. To this end, we also conducted extensive user studies (semi-structured interviews and an online user study) and empirical analyses. Our semi-structured interviews with machine learning practitioners highlighted that 84% of them regularly face explanation disagreements. While there exist metrics and frameworks to both evaluate and compare explanations output by state-of-the-art methods (Agarwal et al., 2021; 2022; Han et al., 2022), several practitioners (86% of respondents in our online user study) admitted to using ad hoc heuristics or being uncertain about resolving explanation disagreements in practice. This highlights a critical gap between research and practice in explainable machine learning.

In this work, we prioritized examining the prevalence of explanation disagreement rather than exploring its underlying causes, given the complexity of this analysis. Nonetheless, subsequent research building

on our work has attempted to understand the sources of these disagreements. For example, (Han et al., 2022) extended our work to demonstrate that state-of-the-art explanation methods all perform local linear approximations of underlying models but do so in different ways (e.g., using different loss functions and over different local neighborhoods). This can be a source of disagreement between the explanation methods. Furthermore, our results raise the question of how to resolve explanation disagreements when they occur. While this remains an active area of research, one approach to selecting among explanation methods is to first define the desired goal of an explanation and then choose the method that best meets this goal. For example, follow-up work by Han et al. (2022) defines local function approximation as the desired goal of an explanation, develops a theoretical framework to analyze how faithfully each method performs local function approximation and advises selecting the method that performs it most faithfully over a given input space. Given the high-stakes applications of model explanations, using a principled and systematic approach to select among explanation methods is preferable to the ad hoc strategies currently employed by practitioners. Finally, the extent of explanation disagreement would also depend on the specific metric being used to measure the disagreement. For instance, while our results broadly demonstrate a moderate to high degree of disagreement among all pairs of explanations across all data modalities, we observe that the extent of disagreement is slightly lower when we consider variants of the proposed metrics. See Appendix D.2 for additional details.

## Acknowledgements

We thank the anonymous reviewers for their insightful comments and feedback on this paper. We also thank Javin Pombra for assisting with preliminary results during the initial phases of this study. Lastly, we are very grateful to Ana Lucic and Maurits J. R. Bleeker for pointing out the omission of a relevant study and helping us improve the contextualization of this work.

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

## A  Measuring Disagreement

In this section, we formally define our 6 different metrics: *feature agreement*, *rank agreement*, *sign agreement*, *signed rank agreement*, *rank correlation*, and *pairwise rank agreement*. The first four metrics capture disagreement for the top-$k$ features of the explanations and the last two metrics capture disagreement for a selected set of features that could be provided as input by an end user. For all six metrics, lower values indicate stronger disagreement.

### A.1  Measuring Disagreement For Top-$k$ Features

We now define four metrics, which capture specific aspects of explanation disagreement for the top-$k$ features.[5]

---

[5]The top-$k$ features of an explanation are typically computed only based on the magnitude of the feature importance values and not the signs.

**Feature Agreement:** ML practitioners in interviews (Section 3.1) indicated that a key notion of disagreement between a pair of explanations is that they output different top-$k$ features. To capture this notion, we introduce the feature agreement metric which computes the fraction of common features between the sets of top-$k$ features of two explanations. Given two explanations $E_a$ and $E_b$, feature agreement is formulated as:

$$FeatureAgreement(E_a, E_b, k) = \frac{|TF(E_a, k) \cap TF(E_b, k)|}{k}$$

where $TF(E, k)$ returns the set of top-$k$ features of explanation $E$ based on the magnitude of the feature importance values. If the sets of top-$k$ features of explanations $E_a$ and $E_b$ contain the same features, then $FeatureAgreement(E_a, E_b, k) = 1$.

**Rank Agreement:** Practitioners also indicated that if the ordering of the top-$k$ features is different for two explanations (even if the feature sets are the same), then they consider it to be a disagreement. To capture this notion, we introduce the rank agreement metric which computes the fraction of features that are not only common between the sets of top-$k$ features of two explanations but also have the same position in the respective rank orders. Rank agreement is a stricter metric than feature agreement since it also considers the ordering of the top-$k$ features. Given two explanations $E_a$ and $E_b$, rank agreement is formulated as:

$$RankAgreement(E_a, E_b, k) = \frac{|\bigcup_{f \in F} \{f \mid f \in TF(E_a, k) \wedge f \in TF(E_b, k) \wedge R(E_a, f) = R(E_b, f)\}|}{k}$$

where $F$ is the complete set of features in the data, $TF(E, k)$ is as defined above, and $R(E, f)$ returns the rank of feature $f$ according to explanation $E$. If the rank-ordered lists of top-$k$ features of explanations $E_a$ and $E_b$ match, then $RankAgreement(E_a, E_b, k) = 1$.

**Sign Agreement:** In our study, practitioners also mentioned that they consider two explanations to disagree if the feature attribution signs (i.e. directions of feature contribution) do not align for the top-$k$ features. To capture this notion, we introduce the sign agreement metric which computes the fraction of features that are not only common between the sets of top-$k$ features of two explanations but also share the same sign in both explanations. Sign agreement is a stricter metric than feature agreement since it also considers signs of the top-$k$ features. Given two explanations $E_a$ and $E_b$, sign agreement is formulated as:

$$SignAgreement(E_a, E_b, k) = \frac{|\bigcup_{f \in F} \{f \mid f \in TF(E_a, k) \wedge f \in TF(E_b, k) \wedge S(E_a, f) = S(E_b, f)\}|}{k}$$

where $F$ and $TF(E, k)$ are as defined above and $S(E, f)$ returns the sign of feature $f$ according to explanation $E$.

**Signed Rank Agreement:** This metric fuses the above three notions of explanation disagreement and computes the fraction of features that are not only common between the sets of top-$k$ features of two explanations but also share the same feature attribution sign and rank in both explanations. The signed rank agreement is the strictest compared to all the aforementioned metrics since it considers both the ordering and the signs of the top-$k$ features. Given two explanations $E_a$ and $E_b$, signed rank agreement is formulated as:

$$SignedRankAgreement(E_a, E_b, k) = \frac{|\bigcup_{f \in F} \{f \mid f \in TF(E_a, k) \wedge f \in TF(E_b, k) \wedge S(E_a, f) = S(E_b, f) \wedge R(E_a, f) = R(E_b, f)\}|}{k}$$

where $F$, $TF(E, k)$, $S(E, f)$, and $R(E, f)$ are as defined above. $SignedRankAgreement(E_a, E_b, k) = 1$ if the top-$k$ features of two explanations match on all aspects (i.e., features, feature attribution signs, rank ordering) barring the exact feature importance values.

### A.2   Measuring Disagreement With Respect to Features of Interest

Practitioners also indicated that they consider two explanations to be different if the relative ordering of features of interest (e.g., salary and credit score discussed in Section 3.1) differ between the two explanations. To formalize this notion, we introduce the two metrics below.

**Rank Correlation:**   We adopt a standard rank correlation metric (i.e., Spearman's rank correlation coefficient) to measure the agreement between feature rankings provided by two explanations for a selected set of features. In practice, this selected set of features corresponds to features that are of interest to end users and can be provided as input by end users. Given two explanations $E_a$ and $E_b$, rank correlation is formulated as:

$$RankCorrelation(E_a, E_b, F) = r_s(R(E_a, F), R(E_b, F))$$

where $F = \{f_1, f_2 \cdots\}$ is a set of features selected by an end user, $r_s$ computes Spearman's rank correlation coefficient, and $R(E, F)$ assigns ranks to features in $F$ based on explanation $E$.

**Pairwise Rank Agreement:**   The pairwise rank agreement takes as input a set of features that are of interest to the user and captures if the relative ordering of every pair of features in that set is the same for both explanations, i.e., if feature A is more important than B according to one explanation, then the same should be true for the other explanation. This metric computes the fraction of feature pairs for which the relative ordering is the same between two explanations. Given two explanations $E_a$ and $E_b$, pairwise rank agreement is formulated as:

$$PairwiseRankAgreement(E_a, E_b, F) = \frac{\sum\limits_{i,j \text{ for } i<j} \mathbb{1}[\text{RO}(E_a, f_i, f_j) = \text{RO}(E_b, f_i, f_j)]}{\binom{|F|}{2}}$$

where $F = \{f_1, f_2 \cdots\}$ is a set of features selected by an end user, the relative ordering function $\text{RO}(E, f_i, f_j)$ is an indicator function that returns 1 if feature $f_i$ is more important than feature $f_j$ according to explanation $E$ and returns 0 otherwise.

## B   Experimental Setup

### B.1   Black Box Models: Training and Performance

For tabular data, we train four models: a logistic regression model, a gradient-boosted tree model (50 estimators), a random forest model (50 estimators), and a densely-connected feed-forward neural network (with 3 hidden layers with relu activation consisting of 50, 100, and 50 neurons, respectively). For the COMPAS dataset, we train the four models based on an 80%-20% train-test split of the dataset, using features to predict the COMPAS risk score group. The test accuracies of the four models are 0.75, 0.79, 0.79, and 0.73, respectively. For the German credit dataset, we train the same four models based on an 80%-20% train-test split of the dataset, using features to predict the credit risk group. The test accuracies of the four models are 0.65, 0.70, 0.73, and 0.64, respectively.

For text data, we trained a widely-used LSTM-based text classifier, based on 120,000 training samples and 7,600 test samples, to predict the news category of the article from which a sentence was obtained. The model performs with 90.67% accuracy. The architecture comprises an embedding layer of dimension 300, followed by an LSTM layer of hidden size 256 connected to a four-dimensional output layer.

For image data, we use the pre-trained ResNet-18 model (He et al., 2016) and analyze explanations generated for predictions made to classify images to one of the 1000 classes. This model performs 69.758 % and 89.078 % on Accuracy@1 and Accuracy@5 metrics[6], respectively.

---

[6]https://pytorch.org/vision/stable/models.html

### B.2 Explanation Methods

For tabular data, the perturbation-based explanation methods (LIME and KernelSHAP) were applied to explain all four models while the gradient-based explanation methods (Vanilla Gradients, Integrated Gradients, Gradient*Input, and SmoothGRAD) were applied to explain the logistic regression and neural network models to explain samples from the test set (1,198 samples for the COMPAS dataset and 200 samples for the German Credit dataset). Because gradients are not computed for tree-based models, the gradient-based explanation methods were not applied to the random forest and gradient-boosted tree models. When applying explanation methods with a sample size hyperparameter (LIME, KernelSHAP, Integrated Gradients, SmoothGRAD), we performed a convergence check and selected the sample size at which an increase in the number of samples does not significantly change the explanations. Change in explanations at the current versus previous sample size is measured by the L2 distance of feature attributions, the L2 distance of the ranks of all features, and the six metrics. For both COMPAS and German Credit datasets, we used 2,000 samples for relevant methods (i.e. LIME, KernelSHAP, SmoothGRAD, and Integrated Gradients).

For text data, we applied all six explanation methods on the LSTM-based classifier to explain predictions for 7,600 samples in the test set. For LIME and KernelSHAP, we follow the convergence analysis described above and find that attributions do not change significantly beyond 500 perturbations; hence, we use 500 perturbations for LIME and KernelSHAP. Integrated Gradients explanations were generated using 500 steps which is higher than the recommended number of steps mentioned in (Sundararajan et al., 2017). SmoothGRAD explanations were generated using 500 samples to get the most confident attribution which is significantly higher than the recommended number of 50 samples (Smilkov et al., 2017).

For image data, we applied all six explanation methods on the ResNet-18 model (He et al., 2016) to explain predictions for the PASCAL VOC 2012 test set of 1,449 samples. Integrated Gradients explanations were generated using 400 steps, significantly higher than the recommendation of 300 (Sundararajan et al., 2017), to obtain a stable and confident attribution map. Similarly, SmoothGRAD explanations were generated using a sample size of 200 which is also higher than the recommended sample size of 50 (Smilkov et al., 2017). For LIME and KernelSHAP, we chose 100 perturbations to train the surrogate model as we did not notice any significant changes in attributions beyond 50 perturbations. KernelSHAP and LIME were used to compute attributions of super-pixels annotated in PASCAL VOC 2012 segmentation maps. Due to a larger feature space in images compared to the previous tabular and text datasets, disagreement metrics based on top-$k$ features may not provide a clear picture. Hence, we use Rank Correlation between attribution maps generated by a pair of explanation methods as the disagreement metric.

## C Results from Empirical Analysis of Disagreement Problem

### C.1 Tabular Data: COMPAS and German Credit Datasets

The tabular datasets consist of the COMPAS and German Credit datasets. The full set of figures showing explanation disagreement for the two datasets, over four models, measured by six metrics, displayed in two formats (metric mean in heatmap and metric distribution in boxplot) for varying values of top-$k$ features can be found in the code repository accompanying this paper. Here, we elaborate on trends of explanation disagreement at the level of metrics, models, and datasets.

At the level of metrics, explanations show stronger disagreement when measured by stricter metrics (such as signed rank agreement) than by less strict metrics (such as feature agreement and sign agreement). Generally, as the number of top-$k$ features increases, metrics show stronger disagreement between explanations. The exception is the feature agreement metric which, by definition equals one (indicating perfect agreement) when the $k$ is the total number of features. For both datasets, the majority of models and explanation pairs have several points for which the rank correlation metric is near zero or even negative, indicating disagreeing, and even opposing, explanations.

At the level of models, explanations for more complex models (such as the neural network) tend to show similar or stronger disagreement than explanations for less complex models (such as the logistic regression model). As discussed in the main section of the paper, as the complexity of the black box model increases,

it may be more difficult to explain the model's decision-making process and more difficult to disentangle the contribution of each feature to the model's prediction. Thus, the higher the model complexity, the more difficult it may be for different explanation methods to generate the true explanation and the more likely it may be for different explanation methods to generate different false explanations, leading to stronger disagreement among explanation methods.

At the level of datasets, explanations for the German Credit dataset showed similar or stronger levels of disagreement than those for the COMPAS dataset. As discussed in the main paper, one possible reason is that the German Credit dataset has more features than the COMPAS dataset, resulting in a larger number of possible ranking and sign combinations assigned by a given explanation method and making it less likely for two explanation methods to produce consistent explanations.

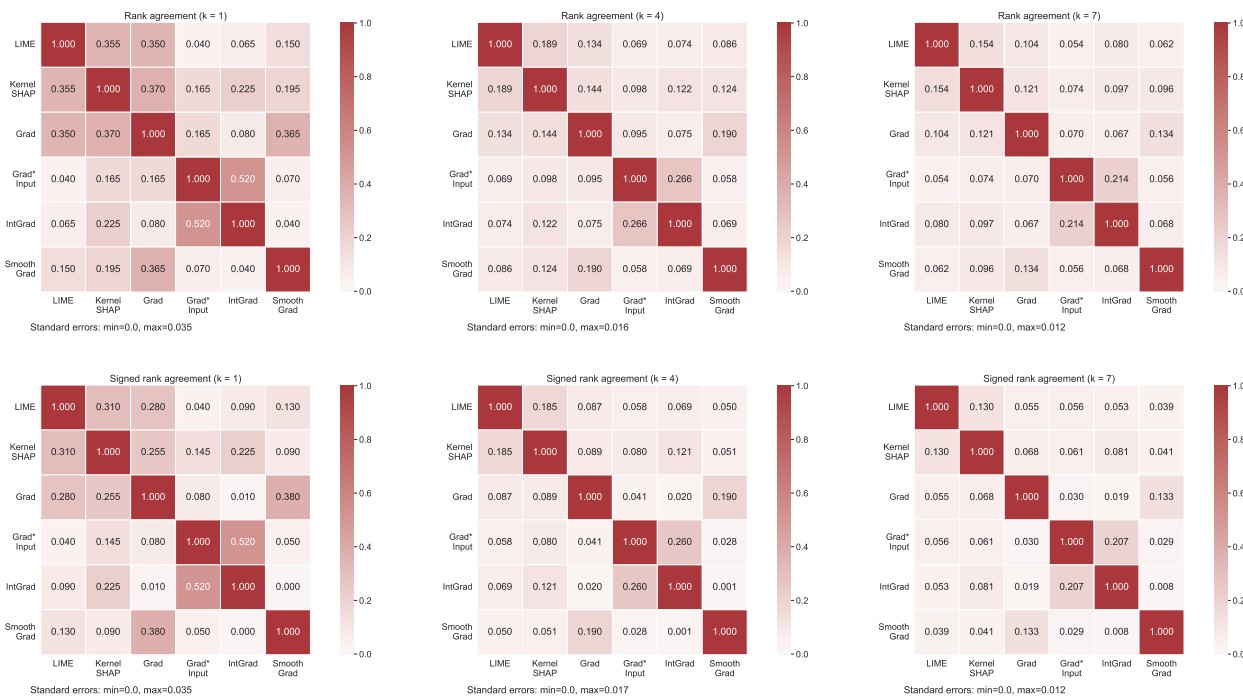

Figure 7: Disagreement between explanation methods for neural network model trained on German Credit dataset measured by rank agreement (top row) and signed rank agreement (bottom row) at top-$k$ features for increasing values of $k$. Each cell in the heatmap shows the metric value averaged over test set data points for each pair of explanation methods, with lower values (lighter colors) indicating stronger disagreement.

## C.2 Text Data: AG_News Dataset

In this section, we analyze the trends in pair-wise disagreement w.r.t the feature agreement metric with the increasing number of features considered for measuring disagreement ($k = [3, 7, 11]$). We observe that agreement between gradient-based methods is significantly greater compared to that with perturbation-based explanation method, which is more prominent for larger values of $k$.

## C.3 Image Data: ImageNet Dataset

In Table 2, we analyze disagreement between perturbation-based explanation methods when feature attributions are computed at the super-pixel level, and observe a significantly high agreement in terms of all the proposed disagreement metrics. However, there is little to no agreement when it's computed at the pixel level in terms of the rank correlation between explanations, shown in Figure 9.

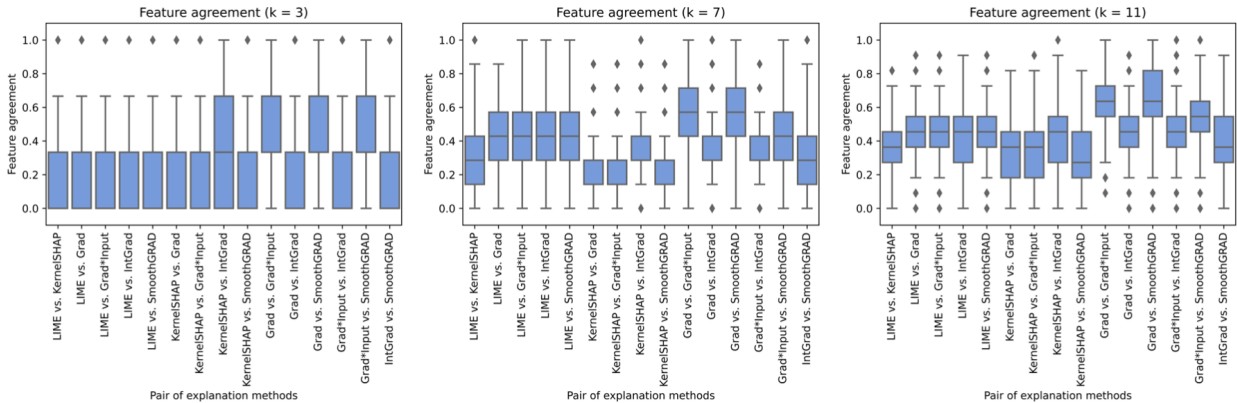

Figure 8: Box plot for feature agreement on AG_News dataset for k = [3,7,11]

| Metrics | ResNet-18 |
|---|---|
| **Rank correlation** | 0.8977 |
| **Pairwise rank agreement** | 0.9302 |
| **Feature agreement** | 0.9535 |
| **Rank agreement** | 0.8478 |
| **Sign agreement** | 0.9218 |
| **Signed rank agreement** | 0.8193 |

Table 2: Disagreement on ImageNet between LIME and KernelSHAP

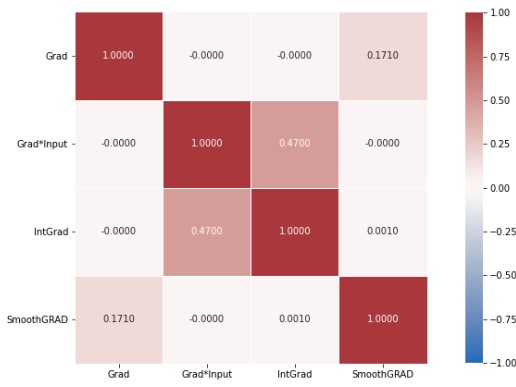

Figure 9: Rank correlation for explanations computed at pixel level by gradient-based explanation methods

# D  Additional Experiments

## D.1  Comparison to L2X (Chen et al., 2018)

For the COMPAS dataset, we show the disagreement between L2X (Chen et al., 2018) and other explanation methods in Figure 10 (neural networks) and Figure 11 (logistic regression). We observe a similar pattern with L2X as well, where there is poor agreement between explanations computed by L2X and those calculated by other explanation methods.

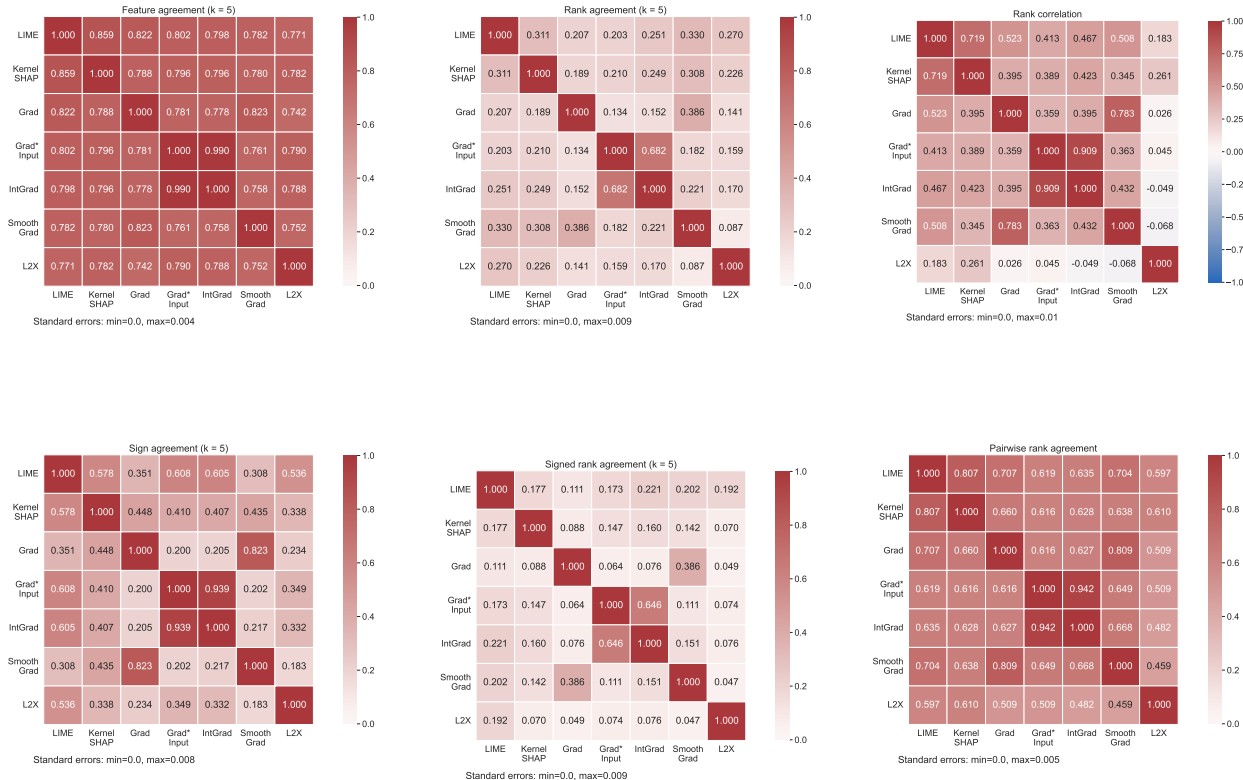

Figure 10: Disagreement between explanation methods for neural network model trained on COMPAS dataset measured by four metrics: feature, rank, sign, and signed rank agreement across top $k = 5$ features.

## D.2  Additional Metrics to Measure Explanation Disagreements

In this section, we study two additional metrics for measuring the disagreement between two explanations.

### D.2.1  Weighted Rank Agreement

Here, we introduce the weighted rank agreement, a new metric that captures the agreement between two explanations. This metric can be intuitively thought of as a softer version of the previously defined rank agreement metric, as it accounts for differences in ranks when computing the agreement between two explanations. Given two explanations $E_a$ and $E_b$, weighted rank agreement is formulated as:

$$WeightedRankAgreement(E_a, E_b, k) = \frac{1}{k} \sum_{f \in TF(E_a, k) \wedge TF(E_b, k)} 1 - \frac{|R(E_a, f) - R(E_b, f)|}{k}$$

where $TF(E, k)$ returns the set of top-k features of explanation E based on the magnitude of the feature importance values, and $R(E, f)$ returns the rank of feature $f$ according to explanation $E$. If the rank-ordered lists of top-k features of explanations $E_a$ and $E_b$ match exactly, then $WeightedRankAgreement(E_a, E_b, k) = 1$. Analogously, if the top-k features of explanations $E_a$ and $E_b$ do not overlap at all (i.e., $TF(E_a, k) \wedge TF(E_b, k)$ is the emoty set), then $WeightedRankAgreement(E_a, E_b, k) = 0$. In all other cases, this metric takes a value between 0 and 1 (excluding both 0 and 1).

We plotted the weighted rank agreement for the COMPAS dataset (See Figure 12) and observed that the degree of agreement is higher according to this metric compared to our original rank agreement metric (See top row, middle column plots in Figures 10 and 11). This makes sense since our original rank agreement was a stricter variant that assigned a value of 1 only when the rank of a feature matched exactly between two

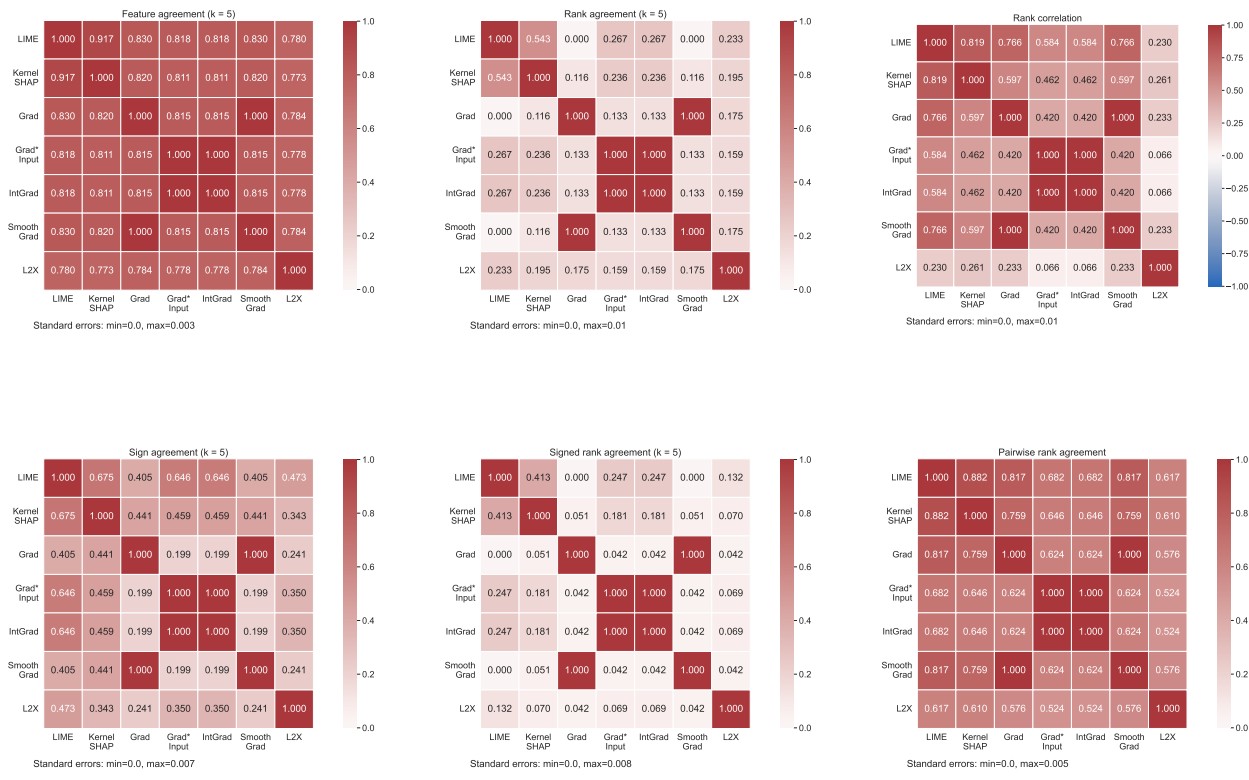

Figure 11: Disagreement between explanation methods for logistic regression model trained on COMPAS dataset measured by four metrics: feature, rank, sign, and signed rank agreement across top $k = 5$ features.

explanations, and assigned a value of 0 otherwise. On the other hand, the new weighted rank agreement is a more lenient version which assigns a value of 1 if there is an exact match in ranks, a value of 0 if the top-k features of the two explanations do not overlap at all, and a value between 0 and 1 to account for differences in ranks in all other cases.

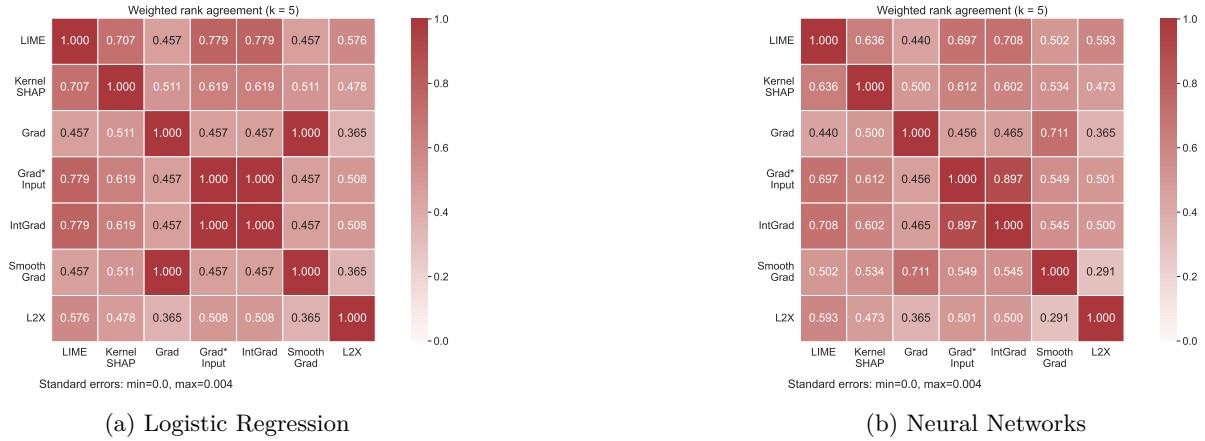

(a) Logistic Regression

(b) Neural Networks

Figure 12: Disagreement between explanation methods for models trained on COMPAS dataset measured using weighted rank agreement metric across top $k = 5$ features.

### D.2.2 Top-K Pairwise Rank Agreement

Here, we introduce a new metric called top-k pairwise rank agreement which is a variant of our previously proposed pairwise rank agreement metric, and it is computed using the top-k features of any two given explanations. Given two explanations $E_a$ and $E_b$, top-k pairwise rank agreement is formulated as:

$$TopKPairwiseRankAgreement(E_a, E_b, F') = \frac{\sum\limits_{f_i, f_j \in F' \text{ and } i < j} \mathbb{1}[\text{RO}(E_a, f_i, f_j) = \text{RO}(E_b, f_i, f_j)]}{\binom{|F'|}{2}}$$

where

$$F' = TF(E_a, k) \cup TF(E_b, k)$$

.

and $TF(E, k)$ returns the set of top-k features of explanation $E$ based on the magnitude of the feature importance values. The relative ordering function $\text{RO}(E, f_i, f_j)$ is an indicator function that returns 1 if feature $f_i$ is more important than feature $f_j$ according to explanation $E$ and returns 0 otherwise.

For the COMPAS dataset, we plot the disagreement between different explanation methods according to top-k pairwise rank agreement in Figures 13 and 14. For both the logistic regression and neural network models, the L2X method exhibits the lowest top-k rank correlation with the other explanation methods compared, suggesting it provides quite different feature importance rankings than the other approaches.

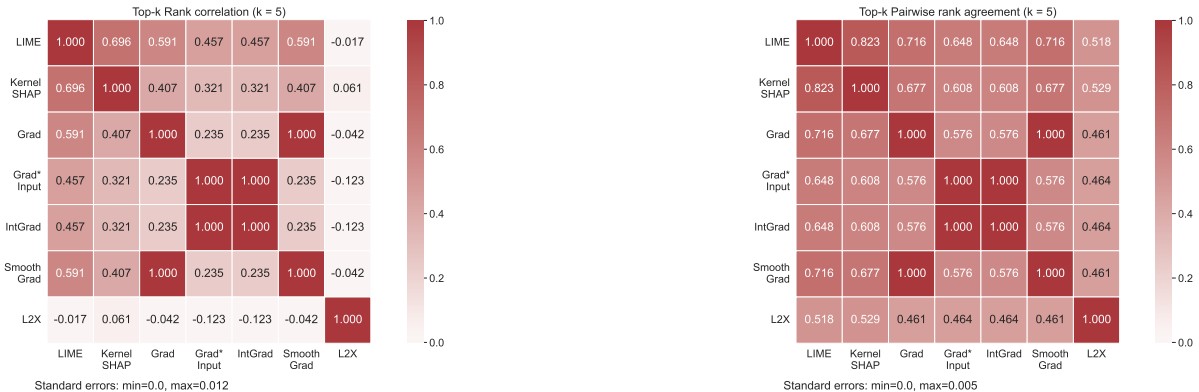

Figure 13: Disagreement between explanation methods for logistic regression model trained on COMPAS dataset measured using Top-K Pairwise Rank Agreement and Top-K Rank Correlation metric across top $k = 5$ features.

### D.2.3 Top-K Rank Correlation

Here, we introduce a new metric called top-k rank correlation which is a variant of our previously proposed rank correlation metric, and it is computed using the top-k features of any two given explanations. Given two explanations $E_a$ and $E_b$, top-k rank correlation is formulated as:

$$TopKRankCorrelation(E_a, E_b, F') = r_s(R(E_a, F'), R(E_b, F'))$$

where

$$F' = TF(E_a, k) \cup TF(E_b, k)$$

As defined previously, $TF(E, k)$ returns the set of top-k features of explanation E based on the magnitude of the feature importance values, $R(E, F')$ is a vector containing the rank of every feature $f \in F'$ according to explanation $E$, and $r_s$ computes Spearman's rank correlation coefficient.

For the COMPAS dataset, we plot the disagreement between different explanation methods according to top-k rank correlation metric in Figures 13 and 14. We observe similar patterns of lower agreement between different explanation methods, with L2X exhibiting the lowest top-k rank correlation with other explanation methods.

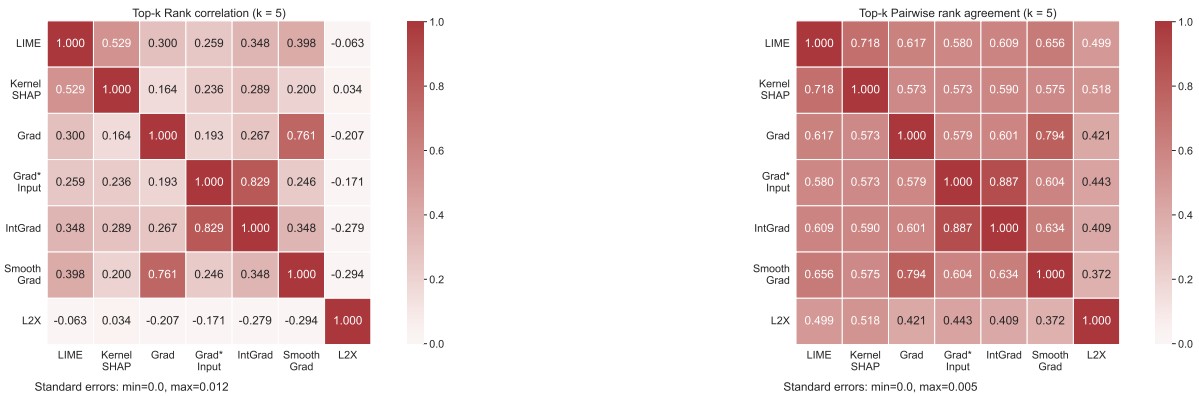

Figure 14: Disagreement between explanation methods for neural network model trained on COMPAS dataset measured using Top-K Pairwise Rank Agreement and Top-K Rank Correlation metric across top $k = 5$ features.

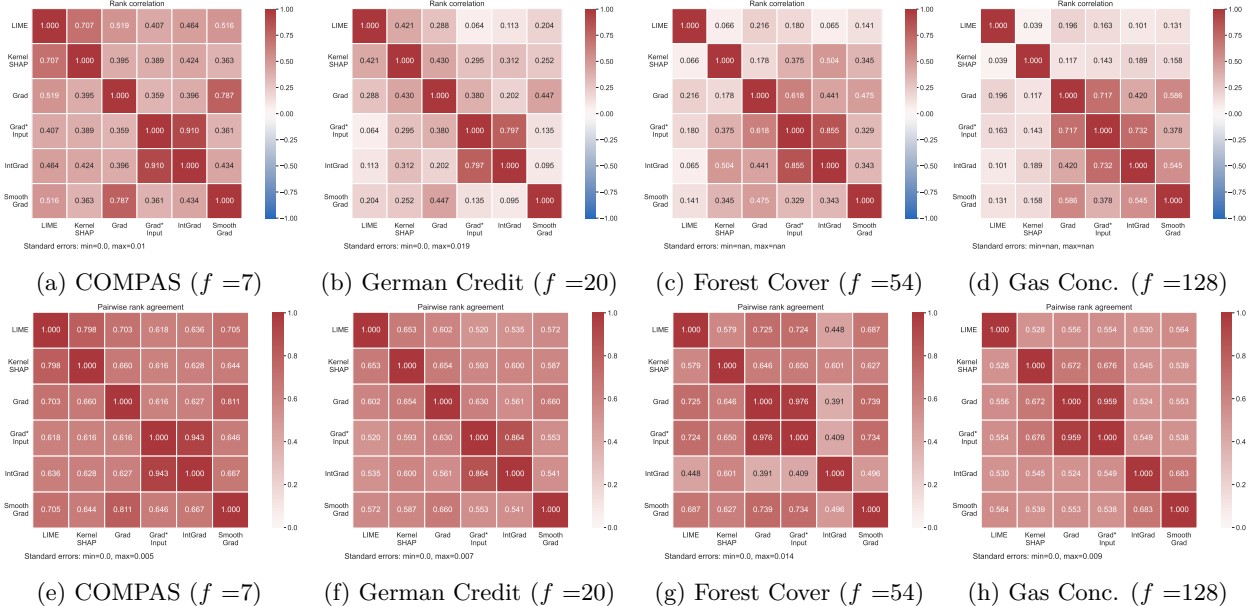

Figure 15: Disagreement in explanation methods for neural network models trained on datasets with the increasing number of features ($f$), as assessed by rank correlation (top row) and pairwise rank agrement (bottom row). Each cell in the heatmap shows the metric value averaged over test set data points for each pair of explanation methods, with lower values (lighter colors) indicating stronger disagreement.

## D.3 Effects of Dataset Complexity

We experimented with two more tabular datasets to analyze the change in disagreement with increasing dataset complexity. These two additional datasets are the Forest Cover Type dataset[7] with 54 features and the Gas Concentration dataset (Rodríguez-Luján et al., 2014) with 128 features. We plot the rank

---

[7]https://archive.ics.uci.edu/dataset/31/covertype

correlation and pairwise correlation agreement metrics in Figure 15 for 4 datasets: COMPAS (7 features), German Credit (20 features), Forest Cover (45 features), and Gas Concentration (128 features) between the 6 explanation methods for neural network models. We did not observe a drastic change in the disagreement between explanations when the features in the tabular data increased from 7 to 128; however, we did notice a substantially lower rank correlation between LIME and KernelSHAP with other methods for the Gas Concentration dataset which has the largest number of features among the 4 datasets.

# E   Omitted Details from Section 5

## E.1   Screenshots of UI

In Figures 16 and 17, we present screenshots of the UI that participants are presented with before beginning the study. The purpose of this introduction page is to familiarize the participants with the COMPAS prediction setting, the six explainability methods we use, and the explainability plots we show in each of the prompts.

## Introduction

COMPAS is a popular commercial algorithm used by judges for determining a criminal defendant's likelihood of reoffending (recidivism).

The COMPAS dataset consists of 7 features:

- **age**
- **two_year_recid**: whether the defendant recidivated within 2 years of the original crime
- **priors_count**: number of prior crimes committed
- **length_of_stay**: length the defendant stayed in jail
- **c_charge_degree**: one of Misdemeanor, Felony
- **sex**: one of Male, Female
- **race**: one of African-American, Asian, Caucasian, Hispanic, Native American, or Other

For this study, we trained a neural network on the COMPAS dataset to **predict a criminal defendant's COMPAS risk score (low or high), corresponding to whether he/she would commit a crime after two years past the date of the original crime**. Since it is important to understand our model's predictions (explainability), we also ran six popular explainability algorithms on various input points.

The explainability algorithms we use are listed here. **You do not need to understand them past what we have described here.**

- **LIME**: an explanation based on a locally linear approximation of the model at that input
- **KernelSHAP**: a combination of LIME and Shapley Values, which identify the contribution of each feature based on interactions with other features
- **Gradient**: the gradient of the model at the input
- **Gradient*Input**: the dot product of the input features and the gradient explanation
- **SmoothGrad**: weighted average of the gradient at points around the input
- **Integrated Gradients**: a modification of the gradient method to satisfy two axioms, *sensitivity* and *implementation invariance*

Figure 16: This is a screenshot of the first half of the introductory page, describing our COMPAS risk score prediction setting and briefly summarizing the six explainability algorithms used (with links to their corresponding papers for the interested participant).

## E.2   Prompts Used

In this section, we share the 15 prompts that we showed users. Each prompt highlights a pair of different explainability algorithms on a COMPAS data point. For each pair, we chose the data point from the entire COMPAS set that maximized the rank correlation between the explanations.

## E.3   User Study Questions

In each of the five prompts, we asked participants the following questions, which we refer to as *Set 1*. Questions 3-4 were only shown if the user selected *Mostly agree*, *Mostly disagree*, or *Completely disagree* to Question (1).

1. To what extent do you think the two explanations shown above agree or disagree with each other? (choice between *Completely agree, Mostly agree, Mostly disagree, Completely disagree*)

2. Please explain why you chose the above answer.

3. Since you believe that the above explanations disagree (to some extent), which explanation would you rely on? (choice between *Algorithm 1 explanation, Algorithm 2 explanation, It depends*)

**Your Task**

On each of the next 5 pages, you will see the result of two explainability methods on the same input sample from COMPAS, as shown below. **Assume that the criminal defendant's risk of recidivism was correctly predicted to be high.** The explanation of the prediction will then be shown to you, as in the figure below.

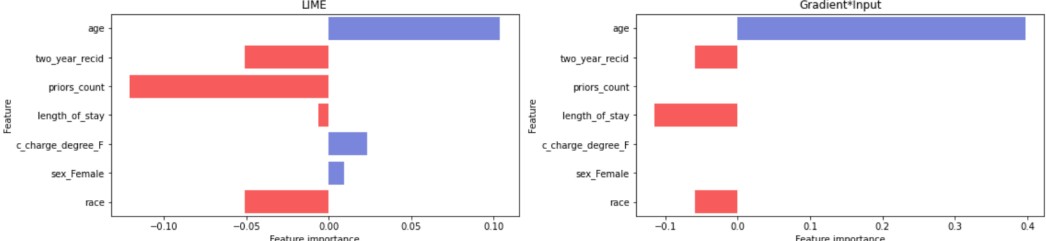

The y-axis lists each of the 7 COMPAS features, and the x-axis shows the importance of that feature. Positive importance values are shown in blue, while negative importance values are shown in red. A **high positive importance** for a feature means that the feature contributed greatly to the correct prediction, while a **high negative importance** means that the feature negatively contributed (was misleading) to the prediction. Note the different x-axis scales resulting from different methods. You will be asked to compare the two explanations.

Figure 17: This is a screenshot of the second half of the introductory page, describing the concrete task and an explanation of what is shown in the explainability plots.

4. Please explain why you chose the above answer.

After answering all five prompts, the user was then asked the following set of questions, which we refer to as *Set 2*. Questions 4-9 were only shown if the user selected *Yes* to Question 3.

1. (Optional) What is your name?

2. What is your occupation? (eg: PhD student, software engineer, etc.)

3. Have you used explainability methods in your work before? (*Yes/No*)

4. What do you use explainability methods for?

5. Which data modalities do you run explainability algorithms on in your day to day workflow? (eg: tabular data, images, language, audio, etc.)

6. Which explainability methods do you use in your day to day workflow? (eg: LIME, KernelSHAP, SmoothGrad, etc.)

7. Which methods do you prefer, and why?

8. Do you observe disagreements between explanations output by state of the art methods in your day to day workflow?

9. How do you resolve such disagreements in your day to day workflow?

**E.4   Further analysis of overall agreement levels**

In this section, we present further plots analyzing responses to questions (1) in Set 1. As shown in Figure 19a, only 32% of responses were *Mostly Agree/Completely Agree* and 68% were *Mostly Disagree/Completely Disagree*, indicating that participants experienced the disagreement problem. We also grouped the responses by prompt, shown in Figure 19b, highlighting that different pairs of algorithms can have different levels of disagreement. We removed prompts with less than 4 total responses. We see that there are varying levels of disagreements among prompts. For example, all participants who were shown the Gradient vs. SmoothGrad prompt believed they agreed to some extent, while all participants who were shown the Gradient vs. Integrated Gradients prompt believed they disagreed to some extent.

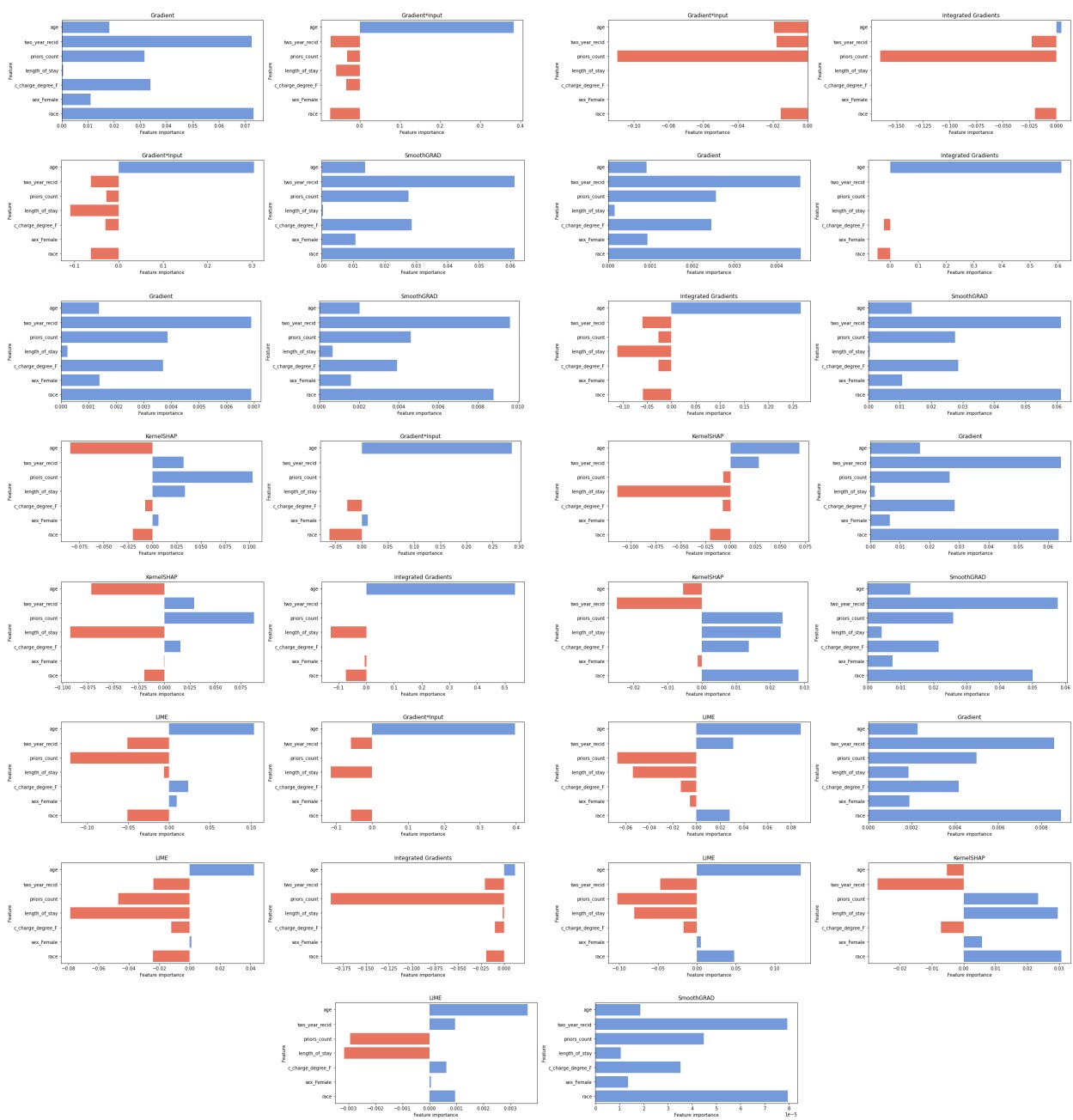

Figure 18: Images showing the 15 prompts we used. Each prompt shows the explanation of the same input point with two different interpretability algorithms.

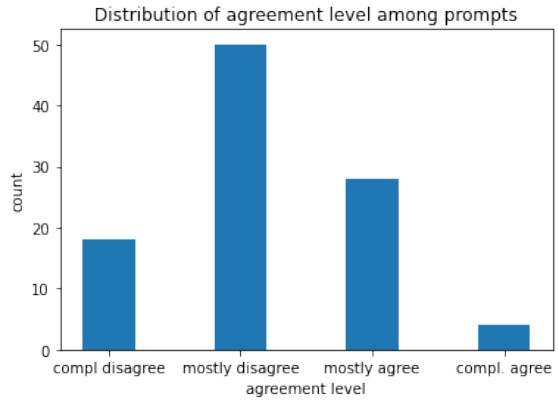

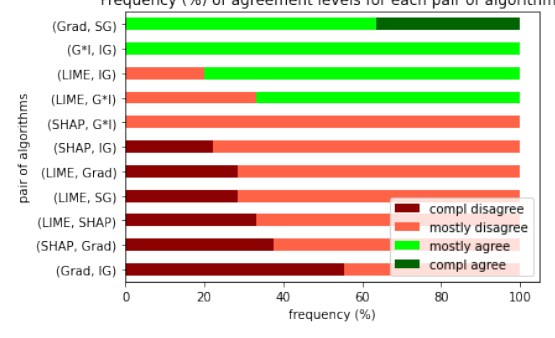

(a) This figure shows the distribution of responses in aggregation over all prompts. The x-axis shows the four possible responses, and the y-axis shows the number of times that response was chosen. Observe that in 68% of cases, participants indicated that the prompts mostly or completely disagreed.

(b) This figure shows the distribution of responses, sorted by prompt. The y-axis shows the pair of explainability algorithms shown in the prompt, and the x-axis shows the frequency that each response was chosen.

Figure 19: These figures show the distribution of answers to Question (1) in Set (1) from Section E.4 in aggregation over all participants.

| Algorithm | Reasons that algorithm was chosen in disagreement |
|---|---|
| **KernelSHAP** | • [36%] SHAP is better for tabular data (*"SHAP is more commonly used [than Gradient] for tabular data"*) 
 • [25%] SHAP is more familiar (*"More information present + more familiarity"*) 
 • [14%] SHAP is a better algorithm overall (*"SHAP seems more methodical than LIME"*, *"SHAP is a more rigorous approach [than LIME] in theory"*) |
| **SmoothGrad** | • [33%] SmoothGrad paper is newer or better (*"SmoothGrad is apparently more robust"*, *"SmoothGrad is often considered improved version of grad"*) 
 • [58%] Reasons based on the explainability map shown (*"directionality of the attributions … [agree] with intuition"*, *"gradient has instability problems [, so] smoothgrad"*) |
| **LIME** | • [54%] LIME is better for tabular data (*"I use LIME for structured data."*) 
 • [15%] LIME is more familiar/easier to interpret (*"I am more familiar with LIME"*, *"LIME is easy to interpret"*) |
| **Integrated Gradients** | • [86%] Integrated Gradients paper is better (*"IG came after gradients and paper shows improvements"*, *"integrated gradients paper showed improvements [over Gradient × Input]"* |

Table 3: Reasons participants chose the top four most favored explainability algorithms (KernelSHAP, SmoothGrad, LIME, and Integrated Gradients) over others when explanations disagreed.

### E.5 Further analysis of reasons participants chose specific algorithms

In this section, we analyze the responses to Set 1, Question (3) in Section E.3. We saw, in 5.2.2, that algorithms such as KernelSHAP were favored over other algorithms. In Table 3, we list the top reasons the four most frequently chosen algorithms were preferred, showcasing direct quotes from participants.

### E.6   Analysis of reasons participants chose neither algorithm

In this section, we analyze the responses to Set 1, Question (4) in Section E.3, focusing on when participants selected *"It depends"* in Question (3), which was chosen in 38% of cases. Again, we present an overarching summary of the reasons participants made this decision in Table 4.

| Rationale | Representative Quote |
|---|---|
| **1. Need more information** | • *"need to see the final prediction of the model and the feature values"* |
| **2. Pick neither explanation** | • *"No compelling reason to choose one over the other. Both don't align with intuition."* |
| **3. Unsure/Don't know** | • *"I'm not sure which of the two methods is more trustworthy"* |
| **4. Would consult an expert** | • *"I would ask a domain expert for his/her opinion"* |
| **5. Combine explanations** | • *"I would combine both – note that age might be doing weird things, but that length of stay and race both contribute to a negative prediction"* |
| **6. Depends on use case** | • *"The two methods have different interpretations - it depends on if I'm more interested in comparing my explanation to some baseline individual state versus just interested in understanding the immediate local behavior"* |

Table 4: Reasons people answered *"It depends"* after being asked to choose between disagreements

### E.7   Further analysis of concluding questionnaire

In this section, we extend the analysis presented in 5.2.3, analyzing the responses to questions in Set 2 of Section E.3. As stated in 5.2.3, we received a total of 20 positive responses to Question (3), but one declined to answer Questions (4) through (9). Therefore, we analyze the remaining 19 responses.

In Question (4), we found that study participants use explainability methods for a variety of reasons such as understanding models, debugging models, help explain models to clients, research. In Question (5), we found that 16 of 19 participants employed explanations for tabular data, 6 of 19 participants for text and language data, 11 of 19 participants for image data, and 1 of 19 for audio data. In Question (6), we found that 14 of 19 participants used LIME, 14 of 19 participants used SHAP, and 13 of 19 participants used some sort of gradient-based methods. Participants also indicated using methods like GradCAM, dimensionality reduction, MAPLE, and rule-based methods. In Question (7), 9 of 19 participants stated that they preferred both LIME and SHAP, with another 3 of 19 participants stating LIME only. We showcase some intriguing answers from Question (7) below:

- "LIME and SHAP seem to be the most universally applicable and I can understand."

- "Methods with underlying theoretical justifications such as KernelSHAP and Integrated Gradients"

- "LIME and SHAP ... [easy to implement] and can work with black box"

- "SHAP and LIME because ... [they are] easy to understand and have standard implementations"

- "LIME, because everything else isn't necessarily capturing what I actually want to know about the local behavior"

Finally, we provide additional quotes highlighting the responses to Questions (8) and (9), which were briefly analyzed in Section 5.2.3. These are shown in Table 5.

### E.8   Breakdown of the Results

The main themes for both academic and industry participants revolve around trusting methods based on their theoretical foundations, intuitive explanations, and suitability for the data type (tabular).

| Category of Response | Samples Quotes |
|---|---|
| **1. Make arbitrary decisions (50%).** | • *"Such disagreements are resolved by data scientists picking their favorite algorithm"*
• *"I try to use rules of thumb based on results in research papers and/or easy to understand outputs."*
• *"I favor LIME and SHAP because there is well documented packages on GitHub"* |
| **2. Unsure/Don't know/Don't resolve (36%)** | • *"there is no clear answer to me. I hope research community can provide some guidance"*
• *"unfortunately there is no good answer at my end … I hope you can help me with finding an answer"* |
| **3. Use other metrics (fidelity) (14%).** | • *"By quantitative assessment of feature importance methods that assess specific properties like faithfulness"*
• *"I might try and use some metric to measure fidelity."* |

Table 5: Representative quotes highlighting themes of how participants address the disagreement problem in their day to day work

In this section, we break down the results by academic (13) and industry (12) participants. For specific pair of explanations, Figure 20 shows that there are some stark differences in how the two groups of participants prefer a specific explanation when there is disagreement between a pair of explanations. For example, when comparing LIME and Integrated Gradients, the academic participants prefer neither while half of the industry participants prefer LIME. In addition, when comparing SHAP and Gradients, the academic participants are split between selecting SHAP or preferring neither of the explanations while the industry participants always prefer SHAP. Figure 21 shows that while both groups prefer SHAP in case of the disagreement, this preference is even more pronounced in industry participants (80%) than to academic participants ( 60%).

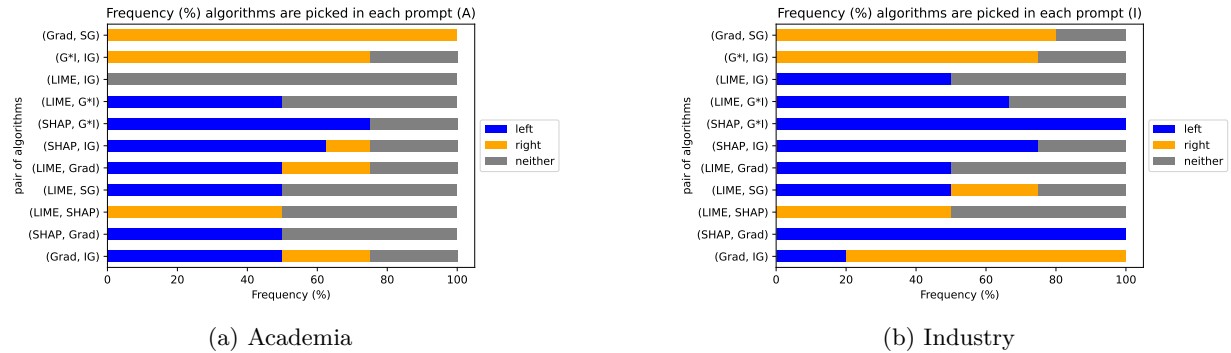

(a) Academia         (b) Industry

Figure 20: The frequency with which each of the explanations was chosen when there is a disagreement. The blue, gold, and grey bars show the percentage of participants (X axis) that picked the left, right, and neither method when presented with the pair of methods shown on the Y axis. (a) is for participants from academia and (b) is for industry participants.

Moreover, industry participants place a stronger emphasis on the theoretical rigor of the methods (42%) compared to academic participants (25%), while academics prioritize explanations aligning with their intuition more often (40%) than industry professionals (26%). A breakdown of these results along with sample quotes from the participants are shown in Tables 7 and 6.

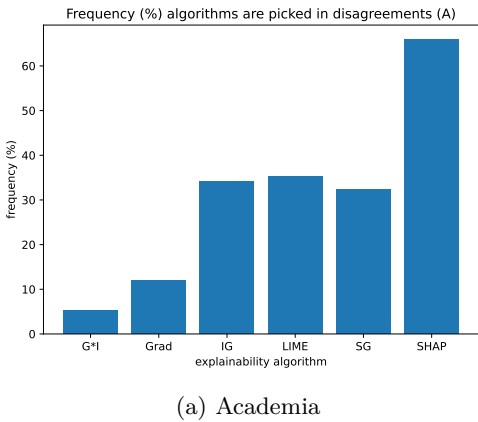
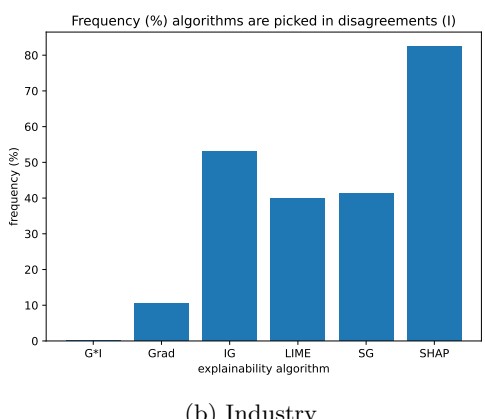

(a) Academia          (b) Industry

Figure 21: The frequency with which each of the explanations was chosen when there is a disagreement. X axis indicates the explanation method and Y axis indicates the frequency. (a) is for participants from academia and (b) is for industry participants.

| Theme Highlighted | Sample Quotes |
|---|---|
| **1. One method's paper/theory suggests that it's inherently better (25%).** | • *"SmoothGrad seems to mainly apply to image processing and not tabular tasks"* |
| **2. One explanation matches intuition better (40%).** | • *"My hunch is that the three features in the middle might be relevant to the prediction. I rely on KernelSHAP because it better aligns with my guess, which is not necessary true."* |
| **3. LIME/SHAP are better for tabular data (7.5%).** | • *"I prefer LIME and SHAP since those are more well motivated and work for tabular data."* |

Table 6: Themes summarizing how **academic** participants decided between explanations when faced with disagreement along with quotes.

| Theme Highlighted | Sample Quotes |
|---|---|
| **1. One method's paper/theory suggests that it's inherently better (42%).** | • *"I pick IG explanation because that paper seems more rigorous and shows improvements over other baselines including gradient\*input"* |
| **2. One explanation matches intuition better (26%).** | • *"The gradient explanation method gives a positive attribution to features that seem intuitively relevant for predicting recidivism (e.g., two_year_recid, priors_count, length_of_stay)."* |
| **3. LIME/SHAP are better for tabular data (26%).** | • *"SHAP is more commonly used for tabular data"* |

Table 7: Themes summarizing how **industry** participants decided between explanations when faced with disagreement along with quotes.

