# OpenReview forum: "The Disagreement Problem in Explainable Machine Learning: A Practitioner’s Perspective"
_TMLR — Accepted by TMLR_

### Review · Reviewer_HUrm · 2024-02-19

**Summary Of Contributions:**

The paper studied the disagreement problem in post-hoc explanations of machine learning models. The authors conducted interviews with data practitioners and obtained insights into the nature of the problem and approaches to dealing with it in practice. The paper also presented a set of metrics to quantify disagreement followed by a quantitative analysis of popular explanation methods in various settings.

**Audience:**

Yes

**Broader Impact Concerns:**

There are no discernible ethical concerns about this paper.

**Claims And Evidence:**

Yes

**Requested Changes:**

Please see the above section.

**Strengths And Weaknesses:**

**Strengths:**

- The paper provided interesting insights into the difficulty of using model explanation methods in practice and potential biases involved in the selection of methods. The qualitative studies with human users provided consensus among data practitioners about the presence of the issue.

- The paper is well-written and cohesive. The research questions and contribution are clear and well-supported by a reasonable experimental setup.

**Weaknesses:**

- The authors should discuss more on the differences in the disagreement patterns among families of methods. For example, LIME uses local linear approximations, SHAP is based on Shapley value while IntGrad, SmoothGrad are gradient-based. This means they use different data signals to measure the “importance” scores and these technical differences should be taken into account when comparing methods.

- The explanation methods under analysis are quite old. There are another line of methods where the explanation model is also a deep learning  model, notably L2X (Chen et al., 2018).
Chen et al. (2018). Learning to Explain: An Information-Theoretic Perspective on Model Interpretation.

- The conclusion from the online user study might make the studied problem less important.

*“One approach to select among explanation methods is to first define the desired goal of an explanation and then select the explanation method that best meets this goal.”*

This means one can in practice ignore the disagreement and select the method that is superior on a metric of interest, such as prediction fidelity as an indicator which one yields the most desirable explanations. This is basically what the research community is proposing, where the proposed method would surpass the baselines on certain desiderata, which also a factor explaining for the disagreement in features. Therefore, it is unclear whether the disagreement is truly a critical issue in practice or it can simply be resolved based on a set of comprehensive metrics.

---

> ### Author Response · Authors · 2024-05-07
> **Response to Reviewer HUrm**
>
> We thank the reviewer for insightful feedback on our work. Please find responses to your questions below.
>
> **Differences between disagreement patterns among the different families of explanations:** Our focus was on highlighting the prevalence of disagreement within the literature on explainable AI and examining how practitioners address these disagreements. To this end, we conducted extensive user studies and empirical analyses. Given the comprehensive nature of our analysis aimed at highlighting the disagreement problem, we prioritized examining the phenomenon of disagreement over exploring its underlying causes. Nonetheless, subsequent research building on our work seeks to understand the sources of these disagreements. For example, [1] builds on our work and demonstrates that the explainability methods we discussed in our paper all perform local linear approximations of underlying models but do so differently (i.e., using different loss functions and over different local neighborhoods). This can be one source of disagreement between the explanation methods. We modified the discussion in Section 6 to reflect this more clearly.
>
> **Comparison to Learning to eXplain (L2X) method:** To address the reviewer’s comments, we conducted additional experiments to incorporate L2X in our analysis. In particular, for the tabular dataset COMPAS, we computed the disagreement of L2X with our initial 6 explanation methods using all of our metrics. We provided results for explanations learned on logistic regression and neural network models. These new results are included in Appendix D.1 (the revision will be uploaded shortly). We observed that, for all metrics, the disagreement between L2X and other explanation methods is at least as high as the disagreement between our initially used explanation methods. We would be happy to include these results in the final version.
>
> **Conclusions from the study:** The main conclusions of our study are as follows: 1) Explanation methods often disagree with each other in terms of the basic insights they convey, and 2) practitioners adopt ad hoc heuristics when resolving explanation disagreements in practice. While we do mention the following in our discussion section: 'One approach to selecting among explanation methods is to first define the desired goal of an explanation and then select the explanation method that best meets this goal,' this statement does not diminish the importance of the studied problem for the following reasons:
> a) Practitioners may not always be clear about the desired goal of an explanation in practice. For example, both fidelity and stability of an explanation could be critical in a real-world application, making it difficult to prioritize one over the other.
> b) Even if the desired goal of an explanation is clear (e.g., fidelity), there are several metrics in the literature that measure fidelity of an explanation (e.g., ROAR [4], PGI, PGU [2]), and recent work has argued that these metrics themselves disagree with each other [3].
> c) Our finding that practitioners were not immediately considering these metrics in a systematic way when faced with explanation disagreements, and instead were adopting ad hoc heuristics to resolve them, reinforces the aforementioned points and emphasizes potential knowledge gaps among practitioners that need to be bridged.
>
>
> **References**
>
> [1] Tessa Han, Suraj Srinivas, and Himabindu Lakkaraju. Which explanation should I choose? A function approximation perspective to characterizing post hoc explanations. Advances in Neural Information Processing Systems (NeurIPS), 2022.
>
> [2] Chirag Agarwal, Eshika Saxena, Satyapriya Krishna, Martin Pawelczyk, Nari Johnson, Isha Puri, Marinka Zitnik, and Himabindu Lakkaraju. OpenXAI: Towards a transparent evaluation of model explanations. Advances in Neural Information Processing Systems (NeurIPS), 2022.
>
> [3] Brian Barr, Noah Fatsi, Leif Hancox-Li1, Peter Richter, Daniel Proano, and Caleb Mok. The Disagreement Problem in Faithfulness Metrics. 2023.
>
> [4] Sara Hooker, Dumitru Erhan, Pieter-Jan Kindermans, Been Kim. A Benchmark for Interpretability Methods in Deep Neural Networks. Advances in Neural Information Processing Systems (NeurIPS), 2019.

---

### Review · Reviewer_jStG · 2024-03-14

**Summary Of Contributions:**

This submission studies disagreements between post hoc explanations provided by different methods for the same prediction. In the first part of the work, interviews are conducted with data scientists to determine the extent of the disagreement problem and what they view as constituting disagreement. Based on the interview findings, six metrics are proposed to quantify disagreement, four defined in terms of top-k features, and two in terms of features of interest to the practitioner. The authors then empirically assess disagreement according to these metrics across six explanation methods and four datasets covering tabular, text, and image modalities. Finally, another user study is performed to characterize how practitioners resolve disagreements between explanations. The most prominent findings are that disagreements between methods often occur, and that practitioners often use heuristics to resolve disagreements.

**Audience:**

Yes

**Claims And Evidence:**

No

**Requested Changes:**

I think it is important to address weaknesses 1 and 2 above (i.e., perform additional analysis or make a best effort to). These are the reason why I have answered "No" for now to the "Claims and Evidence" question, where "No" really means "not always." The inconsistencies should be easy to address. Addressing the minor comments would strengthen the submission.

In addition, I have one question for curiosity:
- Section 4.3.3: Do the authors think that the higher agreement between KernelSHAP and LIME could be due to super-pixels being at a coarser (and less noisy) level of granularity?

**Strengths And Weaknesses:**

## Strengths
- Among the first works to study disagreement between explanation methods (as opposed to inconsistency or instability of a single method)
- I think the combination of human-centered approaches (gathering input from practitioners on what constitutes disagreement and how they resolve it) and more technical approaches (definition of metrics, empirical evaluation across explanation methods and datasets/modalities) is great.
- The paper is written very clearly overall.

## Weaknesses
1. While I do not disagree with the main finding that explanation methods significantly disagree, I wonder whether the degree might be overstated. Specifically:
    1. Rank agreement: This metric defines disagreement as anything other than an exact match in feature ranks. However, a difference in ranks of 1 is better than a difference of say $k$ (so that one of the ranks is not even in the top $k$). What about computing a softer version of rank agreement that takes the difference in ranks into account?
    1. In the experiments (Section 4.2), rank correlation and pairwise rank agreement are computed across all features, whereas in Section 3.2.2, they are intended to be focused on features of interest to the user. I understand that without an actual user, there may be no good way of selecting "features of interest." But using all features may not be reasonable either, because many features are likely to be equally unimportant (have true importance scores near zero) and it is not reasonable or necessary to expect two explanation methods to rank them in the same order. One possibility is to also restrict rank correlation and pairwise rank agreement to the top $k$ for different values of $k$.
1. Conclusions that I think are questionable:
    1. Page 8, "rank agreement and signed rank agreement are lower for the German Credit dataset than for the COMPAS dataset at top 25%, 50%, 75%, and 100% of features": I think a better comparison would be to use the same number $k$ of top features, not the same percentage. As the rest of the paragraph notes, the German Credit dataset has more features so agreement could be lower just because of this reason.
    1. Figure 6b, aggregation of explanation method choices over all prompts: Did the authors adjust for the levels of disagreement in the 15 selected examples? It could be that within this set of 15, the average level of disagreement seen by each method varies considerably. Without adjusting for disagreement level, I am not sure if anything should be inferred from Figure 6b.
1. Inconsistencies:
    1. Abstract and conclusion, "eight predictive models": I count only six, four for the tabular datasets, and one each for text and images.
    1. Page 10, paragraph 2, "Integrated Gradients has the lowest correlation with the rest of the gradient-based methods, this correlation is still significantly higher than its correlation with KernelSHAP and LIME": I do not see this either from Figure 3 middle panel or Figure 4 upper left panel. All of the mean rank correlations for Integrated Gradients are in the same range (high 0.10's to high 0.20's).
    1. Same paragraph, "rank correlation of 0.4-0.6 for LIME as opposed to 0.2-0.4 for KernelSHAP": Again I do not see this from Figure 4 upper left panel.
    1. Section 5.1, "19 participants in this study also participated in the interviews in Section 3.1": As I understand it, all participants in Section 3.1 were from industry. So how can there be only 12 industry participants in Section 5.1?

### Minor comments and questions
- Section 2: How could one substantiate that Han et al. (2022), Banegas-Luna et al. (2023) are "subsequent works"? From the date of the first pre-print version of this submission (which I understand cannot be disclosed yet due to anonymity)?
- Section 3.2: I would consider moving the formal definitions of metrics from Appendix A into this section. They would clarify the discussion and would not take that much space. If space is a concern, the writing in the rest of Section 3 could be condensed as it is a bit repetitive. Also in the metric definitions in Appendix A.1, I think the union over $f \in F$ is unnecessary, it can just be a single set { $f \in F | \dots $ }.
- "Features of interest": The first time that this term is introduced, it could be clarified that they are of interest to the data scientist or domain expert.
- "State-of-the-art" explanation methods: I think this term is repeated too much and could even be omitted, considering that none of the explanation methods evaluated were published later than 2017.
- Figure 1 caption (and subsequent captions), "considered non-trivial if less than 75% ... if rank correlation is less than 0.50": What is the basis for these thresholds? In any case, I do not see them referred to in the main text.
- Page 8 below Figure 2, "wide range of values across explanation method pairs ... when explaining multiple data points": I think this should be rephrased. The point seems to be about variation across data points, not across method pairs. A related question is whether the left panel of Figure 3 is showing the distributions over data points corresponding to the means shown in Figure 1 upper left panel.
- Page 11, paragraph 2, "assume that the criminal defendant's risk of recidivism is correctly predicted to be high risk": Why are all the examples high risk? Are the predictions indeed correct?
- Page 11, paragraph 3, "we chose a different data point from COMPAS to run the two methods, giving us a set of 15 prompts for each of the 15 pairs of explanation methods": This should also be rephrased. Does this mean that each of the 15 pairs is run on a different data point?
- Introduction and conclusion: I think the headline percentages (84% encountered disagreements, 86% resolved heuristically) should be accompanied by the actual numbers of participants (21 of 25, 12 of 14, if I understand correctly). The number of participants is not large, and while I understand it is difficult to recruit more participants with expertise, I think it is better to be forthright about the numbers.

---

> ### Author Response · Authors · 2024-05-08
> **Response to Reviewer jStG**
>
> We thank the reviewer for their insightful feedback on our work. Below, we respond to specific questions and comments raised by the reviewer.
>
> **What about computing a softer version of rank agreement that takes the difference in ranks into account?**
>
> We computed our rank agreement metric based on the inputs obtained from practitioners in our semi-structured interviews. Practitioners considered any deviation in ranks assigned to features as a problem without accounting for exact differences in ranks. Therefore, we adopted the same notion and quantified our rank agreement metric accordingly.
> However, in response to the reviewer’s question, we also computed a softer version of rank agreement that takes the differences in ranks into account. We refer to this metric as the weighted rank agreement (See Appendix D.2.1 for details). We plotted the weighted rank agreement for the COMPAS dataset (Figure 12, Appendix D.2.1) and observed that the degree of agreement is higher according to this metric compared to our original rank agreement metric (See top row, middle column plots in Figures 10 and 11, Appendix D.2.1), as hypothesized by the reviewer. This makes sense since our original rank agreement was a stricter variant that assigned a value of 1 only when the rank of a feature matched exactly between two explanations, and assigned a value of 0 otherwise. On the other hand, the new weighted rank agreement is a more lenient version which assigns a value of 1 if there is an exact match in ranks, a value of 0 if the top-k features of the two explanations do not overlap at all, and a value between 0 and 1 to account for differences in ranks in all other cases.
>
>
> **One possibility is to also restrict rank correlation and pairwise rank agreement to the top k for different values of k.**
>
> In response to the reviewer’s comment, we computed both rank correlation and pairwise rank agreement metrics on the top k features. See Appendices D.2.2 and D.2.3 respectively for details. We plotted the disagreements between different explanation methods using these metrics for the COMPAS dataset. We observe similar patterns of lower agreement scores between different explanation methods as demonstrated in our original results, with the L2X method (an additional explanation technique as suggested by reviewer jStG) exhibiting the lowest agreement with other explanation methods.
>
>
> **a better comparison would be to use the same number k of top features, not the same percentage**
>
> We used the same percentage of top features across datasets to ensure comparability among tabular, text, and image datasets. Text and image datasets are much higher dimensional than the tabular datasets we employ. Therefore, setting k to smaller values would not be very informative for text and image datasets, as this corresponds to just a handful of words in a document or pixels within an image. To ensure consistency across different data modalities, we employed the same percentage of top features instead of the same number of top features. That said, we also compared the same number of top features (k = 1, 4, 7) for the COMPAS (See Figure 2) and the German Credit datasets (see Figure 7 in the updated draft)  and found that the degree of disagreement is higher in case of the German Credit dataset for the same values of k.
>
> **Figure 6b, aggregation of explanation method choices over all prompts: Did the authors adjust for the levels of disagreement in the 15 selected examples?**
>
> Yes, we adjusted for the levels of disagreement in the 15 selected examples. Specifically, we chose these 15 examples as follows: For each pair of explanation methods (6 choose 2 = 15), we selected a data instance with the highest average disagreement across all six of our metrics. This resulted in 15 examples, one for each pair of methods. We also ensured that the degree of disagreement exhibited by the explanations corresponding to all these 15 selected examples is comparable across different metrics.
>
>
> **Eight predictive models in abstract and conclusion**
>
> We fixed this typo and updated the number of models to six in our abstract and conclusion.
>
>
> **Page 10, paragraph 2 (Section 4.3.2), Insights about Integrated Gradients, LIME, and KernelSHAP**
>
> Thanks for pointing out inconsistencies in our findings about Integrated Gradients, LIME, and KernelSHAP in the second paragraph of Section 4.3.2. We edited the text (second paragraph of Section 4.3.2, Page 9 in the updated draft) to correct these errors.

---

> ### Author Response · Authors · 2024-05-08
> **Response to Reviewer jStG (Cont'd)**
>
> **19 participants in this study also participated in the interviews in Section 3.1?**
>
> Thanks for pointing out this error. There is an overlap of 12 participants between our first and second user studies, all of whom are from the industry. We also tried our best to re-recruit the same participants (although some were unavailable) in our second study so that we could closely examine their approaches to resolving explanation disagreements. We corrected the text in the updated draft as follows: ``12 participants in this study also participated in the interviews in Section 3.1.” and also added a footnote.
>
> **Minor Comments:**
>
> We appreciate the other minor comments raised by the reviewer and we will address these in the final version.

---

### Review · Reviewer_Mz9B · 2024-04-29

**Summary Of Contributions:**

This work looks at the problem of disagreements between explainability methods. The authors provide an in-depth study on 6 such methods (LIME, SHAP, Vanilla Gradient, Gradient x Input, SmoothGrad, Integrated Gradients) defining a number of disagreement metrics and quantifying the performance of pairs of methods using these metrics and a series of different types of models and datasets. In addition, they conduct a number of interviews with data scientists on how they use explanation methods, how they view disagreements and how they resolve disagreements.

**Audience:**

Yes

**Claims And Evidence:**

Yes

**Requested Changes:**

None of these are submission critical, but it would just strengthen the work:
* Adding more background on the study participants (happy to have it added in the appendix).
* Splitting the interview results based on tenure, industry vs academia, as well as method familiarity groups.
* Adding more recommendations for how to apply these insights/what's next.
* Showing how the results scale for simple and complex datasets in each category.

**Strengths And Weaknesses:**

Strengths:
* I believe the setup of the paper is really strong. The disagreement problem is well known in the community, but having a thorough analysis of how and what happens when methods disagree in practice is useful to broadcast.
* It was useful and interesting to see the results from interviewing practitioners to get an understanding of how the methods are used in practice, and good to see they defined metrics based on those insights (even if similar metrics have been used in the past to evaluate or compare new interpretability methods against prior work).
* They used a number of models and data types, and provided a thorough result set, as well as an in-depth analysis together with hypotheses for when disagreement seems higher based on model and data characteristics.
* The user studies were well designed to get the necessary insights for the study.
* Presenting the insights based on the day to day workflow is a useful viewpoint, and a good way to present the results as it can lead more easily into "what should be changed in the future" recommendations.

Weaknesses:
* The number of interview participants is relatively low. It's not clear how diverse the participant set is in terms of skills or prior familiarity with the methods - it's mentioned that they were screened for some basic knowledge of ML, but not what the knowledge ranges look like, and what methods they used before.
* Given the split between industry and academia, as well as years of experience and familiarity with methods, it would have been interesting to see the split in the results based on this.
* It would have been interesting to see results on a more complex tabular data set. It's great they started with a simplified setup, but it would have been interesting to see how the findings scale with dataset complexity.
* I would have liked to see an investigation on the limitations of each method as well, if people are aware and if that factors into their choices.
* It would be good to have more recommendations for how to apply the insights surfaced, potentially with per-method examples for application spaces.

---

> ### Author Response · Authors · 2024-05-10
> **Response to Reviewer Mz9B**
>
> We thank the reviewer for their insightful feedback on our work. Below, we respond to specific questions and comments raised by the reviewer.
>
>
> **” Given the split between industry and academia, as well as years of experience and familiarity with methods, it would have been interesting to see the split in the results based on this. ”**
>
> This is a great suggestion. We added a new section (Appendix E.8) to break down the results and insights for participants in Academia and Industry. In Figures 21 and 22 we analyzed how participants in each group favor different explanations when there is a disagreement. Our findings indicate that while the preference is similar across the two groups for many pairs of explanations, there are a few pairs such as SHAP and Gradients or LIME and Integrated Gradients where the preferences are drastically different. Moreover, while both groups prefer SHAP in case of disagreement, this preference is even more pronounced in industry participants (80%) than academic participants (60%).
>
> Additionally, in Tables 6 and 7, we analyzed the main theme for participant’s decisions when facing disagreement.  These themes for both academic and industry participants revolve around trusting methods based on their theoretical foundations, intuitive explanations, and suitability for the data type (tabular). However, industry participants place a stronger emphasis on the theoretical rigor of the methods (42 %) compared to academic participants (25 %), while academics prioritize explanations aligning with their intuition more often (40 %) than industry professionals (26 %)
>
>
> **” It would have been interesting to see results on a more complex tabular data set. It's great they started with a simplified setup, but it would have been interesting to see how the findings scale with dataset complexity.”**
>
> In response to the reviewer's comments, we conducted experiments on two more tabular datasets with more features, i.e., the Forest Cover Type dataset (54 features) and the Gas Concentration dataset (128 features). For the COMPAS dataset, we plotted the rank correlation and pairwise rank agreement of pairs of explanations for neural network models that are trained on these datasets (see Figure 15 from Appendix D.3). Broadly, we didn't observe significant changes in the overall pattern of disagreement among all the methods but we noticed a substantially worse rank correlation for LIME and Kernel SHAP with other methods for the Gas Concentration dataset which has the largest number of features.
>
>
> **” I would have liked to see an investigation on the limitations of each method as well, if people are aware and if that factors into their choices. ”**
>
> The participants seem to have focused more on the strengths of each method rather than its limitations. For example, a common theme was "tabular data, so SHAP” or "tabular, so LIME” when preferring LIME and SHAP for tabular data over other gradient-based methods. Another common theme is to prefer Integrated Gradients over other gradient-based methods due to the chronology of publication dates: ``IG is a more recent paper than gradients which improves on gradients.”.
>
> A notable exception is Vanilla Gradients which was overlooked compared to other gradient-based techniques due to its tendency to generate noisy explanations: ``gradient has unstability problems. So, smoothgrad it is”. We would be happy to include a discussion with more details in the final version.
>
> **“ The number of interview participants is relatively low. It's not clear how diverse the participant set is in terms of skills or prior familiarity with the methods - it's mentioned that they were screened for some basic knowledge of ML, but not what the knowledge ranges look like, and what methods they used before. ”**
>
> We acknowledge that a shortcoming of our study is the small number of participants.  A sample of questions asked from participants is provided in Appendix E.3. A breakdown of the number of participants who used explainability techniques are as follows: SHAP (13 participants), LIME (14 participants), Gradient-based methods (13 participants), None (5 participants). We would be happy to include a more detailed analysis of participants’ expertise in the final version.
>
>
> **” It would be good to have more recommendations for how to apply the insights surfaced, potentially with per-method examples for application spaces. ”**
>
> One of our main findings implies that practitioners adopt ad hoc heuristics when resolving explanation disagreements in practice and lack a principled approach to resolve disagreements (see also our response to reviewer HUrm). A main insight of our study, hence, is the need to bridge this knowledge gap.

---

### Comment · Action_Editor_bdGi · 2024-03-06
**Dual submission concerns**

Dear authors,

This paper has been raised as being a potential dual submission to TMLR and FAccT 2024. Please note that TMLR's dual submission
 policy clearly states: "Unlike many other journals, **TMLR only accepts original contributions that don’t reuse the authors’ own prior work**. In particular, we do not accept submissions that are expanded versions of conference papers. "

Please review the dual submission policy (https://jmlr.org/tmlr/editorial-policies.html) and confirm whether this work is under consideration at other venues or not.

Thank you,

Best regards,

AE

---

### Decision · Action_Editor_bdGi · 2024-05-30

**Recommendation:** Accept with minor revision

**Comment:**

I encourage the authors to revise their manuscript based on the reviewers' comments, adding an overall description of the changes made as a comment. I believe the inclusion of the more lenient version of the ranking metric is reasonable, and it should be clearly mentioned in the text how variable the conclusions may be based on which metrics are used for the comparison. This is especially true given that multiple choices may be practical rather than theoretical (e.g. selecting proportions rather than top-k numbers to compare across datasets). For instance, mentioning clearly that a perfect match is expected is important to put the findings in perspective.

**Audience:**

All reviewers agreed that the findings are interesting, despite some disagreement on timeliness.

**Claims And Evidence:**

All reviewers supported the paper's claims and evidence, although one reviewer has noted inconsistencies that have not been fully addressed by the authors.